# A high-throughput neurohistological pipeline for brain-wide mesoscale connectivity mapping of the common marmoset

Meng Kuan Lin[1]*, Yeonsook Shin Takahashi[1]*, Bing-Xing Huo[1], Mitsutoshi Hanada[1], Jaimi Nagashima[1], Junichi Hata[1], Alexander S Tolpygo[2], Keerthi Ram[3], Brian C Lee[4], Michael I Miller[4], Marcello GP Rosa[5,6], Erika Sasaki[7], Atsushi Iriki[8], Hideyuki Okano[1,9], Partha Mitra[1,2]

[1]Laboratory for Marmoset Neural Architecture, RIKEN Center for Brain Science, Wako, Japan; [2]Cold Spring Harbor Laboratory, Cold Spring Harbor, United States; [3]Indian Institute of Technologies, Madras, India; [4]Center for Imaging Science, Johns Hopkins University, Maryland, United States; [5]Department of Physiology and Biomedicine, Discovery Institute, Monash University, Melbourne, Australia; [6]Australian Research Council Centre of Excellence for Integrative Brain Function, Clayton, Australia; [7]Central Institute for Experimental Animals, Kawasaki, Japan; [8]Laboratory for Symbolic Cognitive Development, RIKEN Center for Brain Science, Wako, Japan; [9]Department of Physiology, Keio University School of Medicine, Tokyo, Japan

*For correspondence:
mengkuan.lin@riken.jp (MKL);
yeonsook.takahashi@riken.jp (YST)

Competing interests: The authors declare that no competing interests exist.

**Abstract** Understanding the connectivity architecture of entire vertebrate brains is a fundamental but difficult task. Here we present an integrated neuro-histological pipeline as well as a grid-based tracer injection strategy for systematic mesoscale connectivity mapping in the common marmoset (*Callithrix jacchus*). Individual brains are sectioned into ~1700 20 µm sections using the tape transfer technique, permitting high quality 3D reconstruction of a series of histochemical stains (Nissl, myelin) interleaved with tracer labeled sections. Systematic in-vivo MRI of the individual animals facilitates injection placement into reference-atlas defined anatomical compartments. Further, by combining the resulting 3D volumes, containing informative cytoarchitectonic markers, with in-vivo and ex-vivo MRI, and using an integrated computational pipeline, we are able to accurately map individual brains into a common reference atlas despite the significant individual variation. This approach will facilitate the systematic assembly of a mesoscale connectivity matrix together with unprecedented 3D reconstructions of brain-wide projection patterns in a primate brain.
DOI: https://doi.org/10.7554/eLife.40042.001

## Introduction

The connectional architecture of the brain underlies all the nervous system functions, yet our knowledge of detailed brain neural connectivity falls largely behind genomics and behavioral studies in humans and in model research species such as rodents (*Bohland et al., 2009*). To fill this critical gap, a coherent approach for the mapping of whole-brain neural circuits at the mesoscale using standardized methodology was proposed in 2009 (*Bohland et al., 2009*). Since then, several systematic brain connectivity mapping projects for the mouse have been initialized and established,

including the Mouse Brain Architecture Project (*Pinskiy et al., 2015*) (www.brainarchitecture.org), the Allen Mouse Brain Connectivity Atlas (*Oh et al., 2014*) (connectivity.brain-map.org), and the Mouse Connectome Project (www.mouseconnectome.org) (*Zingg et al., 2014*). Non-human primates (NHPs) were also proposed as an important group in which to study whole-brain neural architecture. However, the high-throughput experimental approaches for mouse do not automatically apply to NHPs due to bioethical as well as experimental considerations, larger brain sizes coupled with stringent limitations on the numbers, as well as limitations arising from the increased individual variability of the brains.

There has been an increase in the usage of the common marmoset (*Callithrix jacchus*) as a model organism in contemporary neuroscience research (*Izpisua Belmonte et al., 2015*; *Kishi et al., 2014*; *Miller et al., 2016*; *Okano and Kishi, 2018*; *Okano et al., 2016*) (*Figure 1—figure supplement 1*). Marmosets offer a number of experimental advantages over the macaque, including lower cost, ease of handling and breeding (*Kishi et al., 2014*; *Okano and Mitra, 2015*), smaller brain sizes (≈35 mm*25 mm*20 mm) that potentially allow more comprehensive analysis of the neuronal circuitry, and importantly the development of transgenic marmosets and the application of modern molecular tools (*Park et al., 2016*; *Sasaki et al., 2009*; *Sato et al., 2016*).

Marmosets are New World monkeys, in contrast with the Old World macaque monkeys which are the pre-eminent NHP models used in basic and pre-clinical neuroscience research. As depicted in *Figure 1a*, New World monkeys, together with Old World monkeys, apes and humans, form the simian primates (order Primates, infraorder Simiiformes). Simians diverged from prosimians such as lemurs and lorises approximately 85 million years ago (Mya). Among the simians, New World monkeys have evolved in isolation from Old World monkeys, apes and humans for at least 40 million years. *Prima facie* this seems to indicate a relative weakness in using marmosets as NHP models in contrast with the macaques. Nevertheless, a good case can be made for marmosets as NHP models of humans, despite the earlier evolutionary divergence.

Marmosets exhibit more developed social behavior (*Miller et al., 2016*) and vocal communication (*Marx, 2016*) traits, thus social-vocal human traits (and corresponding dysfunctions) might be better modeled in marmosets than in macaques. Marmoset brains are smaller than macaque brains and are comparable in size to some rodents (cf. squirrels and capybara, both species of rodents, have brain volumes comparable to marmosets and macaques). However marmosets are phylogenetically closer to humans than rodents, and thus have more commonality in terms of brain architecture (proportionately larger and more differentiated higher order cortical areas, as opposed to primary cortical areas (*Krubitzer and Dooley, 2013*) (*Figure 1*).

Following the BRAIN (Brain Research through Advancing Innovative Neurotechnologies) Initiative in the U.S. and the HBP (Human Brain Project) in Europe in 2013, Japan launched the Brain/MINDS project (Brain Mapping by Integrated Neurotechnologies of Disease Studies) with a focus on the common marmoset (*Callithrix jacchus*) as an NHP model (*Okano and Mitra, 2015*) (http://www.brainminds.jp/). As part of Brain/MINDS, a combined histological/computational pipeline was established at RIKEN to develop a mesoscopic whole-brain connectivity map in the marmoset. The corresponding methodology is described in this manuscript.

Tract-tracing methods remain the gold standard for studying neural circuit structure at the whole brain level (*Bakker et al., 2012*). Previous brain-wide connectivity mapping for non-human primates have been based on literature curation and meta-analyses. A pioneering survey by *Felleman and Van Essen, 1991* reviewed 52 studies, including both anterograde and retrograde tracing results, to generate a connectivity matrix of 33 brain regions in the visual system of macaque monkeys (*Table 1*). Building upon *Felleman and Van Essen (1991)*, a more comprehensive database of macaque brain connectivity, CoCoMac (Collation of Connectivity data on the macaque brain, cocomac.g-node.org) (*Bakker et al., 2012*; *Kötter, 2004*; *Stephan et al., 2001*), surveyed over 400 tracing studies with ~3300 injections and established a connectivity matrix of 58 brain regions (*Modha and Singh, 2010*; *Stephan, 2013*) (*Table 1*). While the historical tracing studies mostly contain qualitative information, more recent studies have aimed at building a quantitative connectivity database of the macaque brain (*Falchier et al., 2002*; *Markov et al., 2014*; *Markov et al., 2011*) (core-nets.org; *Table 1*).

For the marmoset, an online database of >140 retrograde tracer injection studies in about 50 cortical areas is available online (http://monash.marmoset.brainarchitecture.org) (*Majka et al., 2016*). By surveying 35 tract tracing studies (*Supplementary file 2*) in marmosets since the 1970s,

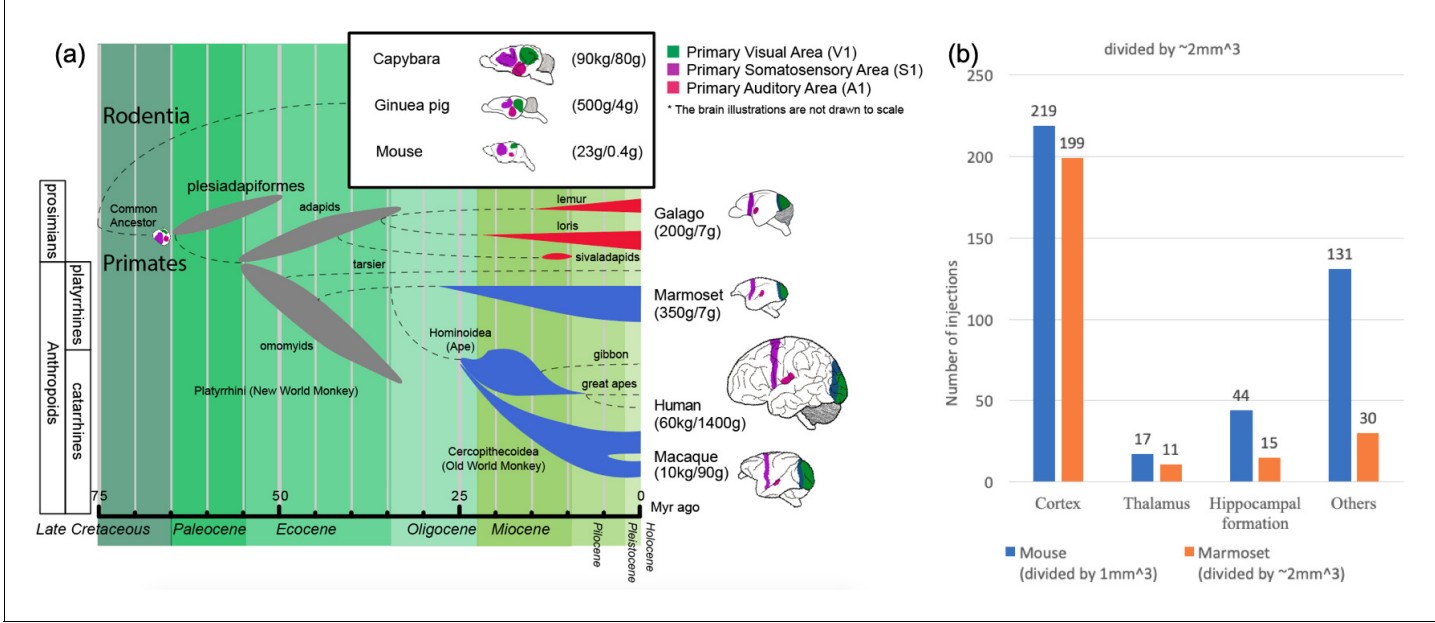

**Figure 1.** Phylogenetic tree of extinct and extant primates and numbers of injection sites achieved (in grid-based tracer mapping) for mouse and marmoset. (a) Phylogenetic tree (*Benton et al., 2009*; *dos Reis et al., 2014*; *dos Reis et al., 2012*; *Janecka et al., 2007*; *O'Leary et al., 2013*; *Mitchell and Leopold, 2015*; *Springer et al., 2011*; *Wilkinson et al., 2011*) showing the ancestral history of extinct and extant primates, after divergence from the common ancestor with rodents (top right inset box) *ca.* 75 million years (Myr) ago. The bottom bar shows geological eras. Thickness of spindle shaped areas in the evolutionary tree indicate prosperity (estimated population and numbers of species) of the group along the history in extinct (gray) prosimian (red) and simian (blue) primates. Each bifurcation represents the species divergence, although the divergence time typically has a wide range and remains controversial. Primates diverged into platyrrhini, the New World Monkey, and catarrini, around 38.9–56.5 million years ago. Catarrini further evolved into Ape, including humans, and Old World Monkey as well as macaque monkeys 25.1–37.7 million years ago. Sketches of the brain in each species are shown on the right, next to their species name. The colored areas in the various brain illustrations indicate the primary visual area as green, somatosensory as purple, and auditory areas as red; each represents an extant primate (bottom right row) and rodent (top inset box) species' body weight (first numbers in brackets) and brain weight (last numbers in brackets) sizes (*Buckner and Krienen, 2013*; *Krubitzer and Dooley, 2013*; *Krubitzer and Seelke, 2012*). Phylogenetic tree adapted from Masanaru Takai (*Takai, 2002*). (b) Fractional brain region volumes, and numbers of injection sites used in grid- based injection plans for marmoset (*Woodward et al., 2018*) and mouse (*Allen institute for brain science, 2017*). Bar plots show the number of grid-injection sites within the displayed compartment in each species, assuming a spacing between injection sites of ~1 mm isometric in mice, and ~2–3 mm isometric in marmosets.

DOI: https://doi.org/10.7554/eLife.40042.002

The following figure supplement is available for figure 1:

**Figure supplement 1.** Number of research articles comparing mouse, macaque and marmoset in 1980-2017.

DOI: https://doi.org/10.7554/eLife.40042.003

we have collected data from over 400 injections, but much of this knowledge cannot be easily integrated with current efforts given the use of older nomenclatures, and the lack of access to primary data. A full connectivity matrix is yet to be established (*Table 1*). Nevertheless existing knowledge about the marmoset visual, auditory, and motor systems indicate strong similarities between marmoset and macaque brain circuitry, suggesting a preserved brain connectivity plan across primates (*Bakola et al., 2015*; *de la Mothe et al., 2012*; *Solomon and Rosa, 2014*). Comparing two NHP brain architectures (marmoset, macaque) will help to better contextualize human brain circuit architecture.

None of these earlier studies in NHPs have used a single, consistent methodology employing a unified experimental-computational workflow, dedicated to systematic mesoscale connectivity mapping. In addition, an automated throughput image analysis is required for the whole-brain circuit reconstruction and mapping (*Hua et al., 2015*). This became the goal of the pipeline described in this paper. Importantly, brain-wide data sets are already available for grid-based tracer mapping projects in the mouse. A corresponding data set generated using similar techniques will allow us to gain a more unified view of primate brain connectivity architecture, and also permit an unprecedented comparative analysis of mesoscale connectivity in rodents and primates.

**Table 1.** Past and present summary of historical tract-tracing studies in macaque and marmoset monkeys.

Three resources of macaque monkey brain connectivity are shown. Felleman and Van Essen (*Felleman and Van Essen, 1991*) and CoCoMac each surveyed a set of studies to generate the connectivity matrix (full reference list in *Supplementary file 2*). Note that CoCoMac is inclusive of the work collected in Felleman and Van Essen (*Felleman and Van Essen, 1991*). Around 235 injections lack tracer direction information. *Markov et al. (2014)* was a single study using only the retrograde tracer to generate the connectivity matrix as well as quantifying the connection strengths. We have surveyed 35 marmoset brain tracing studies that contain 428 tracer injections including both anterograde and retrograde tracers. A complete connectivity matrix is not yet available for the marmoset brain. To date, the most comprehensive marmoset brain connectivity resource available online (http://monash.marmoset.brainarchitecture.org) includes 143 retrograde tracing studies. As part of the current pipeline, we have placed over 188 tracer injections including both anterograde and retrograde tracers. For both macaque and marmoset brain injections, bidirectional tracer injections were double counted as one anterograde and one retrograde tracer injection.

| | Data | Species | Injections | Anterograde tracer | Retrograde tracer | Connectivity matrix | Source |
|---|---|---|---|---|---|---|---|
| Journal papers | No whole-brain image data | Macaque | 370 | 153 | 217 | 33 × 33 | *Felleman and Van Essen, 1991* (52 studies) |
| | | | 3279 | 1429 | 1873 | 58 × 58 | CoCoMac (459 studies) |
| | | | 39 | 0 | 39 | 29 × 91 | *Markov et al., 2014* |
| | | Marmoset | 428 | 93 | 395 | - | 35 studies (Bibliography in supplement) |
| Whole-brain image data | Nissl images overlaid with cell locations (Rosa Lab data set) | Marmoset | 143 | 0 | 143 | - | Online |
| | This paper: Whole-brain set of cross-modal serial sections (Nissl,Myelin, IHC, Fluoro)+MRI | | 188 | 94 | 94 | - | This paper |

DOI: https://doi.org/10.7554/eLife.40042.004

## The injection-grid approach to whole-brain mesoscale connectivity mapping

Mapping the brain-wide neural circuitry in large vertebrate brains remains one of the most important tasks in neuroscience, yet raises tremendous practical and theoretical challenges. The ideal data set would contain the position, morphology, synaptic connectivity together with transmitter/receptor identities at each synapse, and also spatial maps of the diffuse neuromodulatory transmitters and receptors of every neuron. This is clearly not achievable in practical terms. For example, EM based mapping of individual synaptic connectivity and morphology of every neuron remains impractical for a brain as large as the marmoset.

Even if comprehensive mapping was performed in one brain, there would remain the problem of individual variation across brains, which would ideally require doing the same detailed map for many brains. All current approaches to this problem therefore constitute practical compromises (e.g. EM mapping of synaptic connectivity for larger vertebrate brains is currently confined to small brain regions). The grid-injection based approach achieves brain-wide coverage but sacrifices the detailed synaptic connectivity, revealing a species-specific, coarse-grained circuit architecture. The availability of 3D volumetric data sets at light microscopic resolution, with the possibility of quantitative analysis and across-brain comparisons, sets this approach apart from classical neuroanatomical studies which are more targeted (e.g. to individual brain regions for injection placement, possibly to test specific hypotheses) and have largely been carried out in the era before digitizing whole brains was practical.

Within the broad approach, some questions need to be addressed: treatment of individual variation across brains, relation to classical neuroanatomical approaches based on atlas-parcellations, and technical sources of variation, being the difficulty in controlling the locations and sizes of injections, and most importantly the total number of injections. We briefly comment on these inter-related considerations here as they pertain to the design of our injection grid-plan. In the later discussion section, we present some analysis of the degree of individual variation in the data set gathered for this

project, and considerations related to completing whole-brain coverage. Further information may be found in Appendix 9 and 10.

## Planning the grid

Classical neuroanatomical reference atlases list hundreds of individual gray-matter regions or cell groups (including cortical regions and subcortical nuclei), separated by more or less well-defined boundaries. Within regions, continuous gradients may be present. These atlases were developed largely based on the spatial distributions of morphologies and chemo-architectures of the neuronal somata, and to a lesser extent on the connection architecture. As new information becomes available from modern techniques, these atlases are likely to change, also the atlases do not provide *prima facie* information about individual variation, as they are based on an individual brain (or more recently on averages across brains). It is important to take into account the accumulated knowledge represented by these atlases in planning a grid; on the other hand, the atlases themselves represent imperfect knowledge, and sampling brain-space on a regular grid could itself reveal the necessary meso-architecture.

We adopt a compromise, by starting from a roughly regular grid, working backwards from the total number of injections that can realistically be placed/processed within a practical time frame (of several years) and within the constraint of the availability of experimental animals. We therefore started with a grid spacing of ~2 mm, but then adapted the grid in the following ways: (i) grid points overlapping with atlas boundaries were moved to be closer to compartment centers; (ii) atlas compartments smaller than 8 mm$^3$ were assigned injections upto a size cutoff. Placing this size cutoff at 0.27 mm$^3$ produces a total of 356 injection centers in 241 target structures in one hemisphere's grey matter. In cerebral cortex, this corresponds to 221 injection centers in 118 target structures, comprising 74% of the total grey matter volume. Details are presented in Appendix 9.

We inject each site with an anterograde and a retrograde tracer (in separate animals). To maximize utilization of animals we place four injections/animal, 2 anterograde and two retrograde. Our approach is conservative: better availability and utilization of colors in the tracers could permit more injections per animal. Notably, we are able to process significantly more injections per animal than is possible with single-color 2-photon light microscopy, which is important for a primate species such as the marmoset to minimize the number of animals used.

## Individual variation

Classical neuroanatomical studies may place multiple injections in separate animals at a single target to address biological variation. This is impractical for the current approach, it would require too many animals. Nevertheless, we achieve an effective N = 2 per long range projection when combining the results of anterograde and retrograde tracing. Additionally, we tailor injections to the individual variations in animals when using in-vivo MRI guidance to target specific sub-cortical nuclei and using landmarks in injecting cortical sites. Finally, results from different animals are mapped onto a common reference atlas using diffeomorphic mapping utilizing the cytoarchitectonic contrast present in the multimodal histological data gathered in the pipeline. In these ways the grid-approach addresses the issues of individual variation. An analysis of brain compartment size variations across animals, as well as of the injection-size variations, is presented in Appendix 10.

## Materials and methods

A high throughput neurohistological pipeline was established at the RIKEN Center for Brain Science, based on the pipeline developed for the MBA project (*Pinskiy et al., 2015*) at CSHL. The pipeline employed a customized tape transfer assisted cryo-sectioning technique to preserve the geometry of individual sections. Each brain was sectioned serially into a successive series of four 20 μm sections: a Nissl stained section, a silver (Gallyas) myelin stained section, a section stained (ABC-DAB) for the injected cholera toxin subunit B (CTB) tracer and an unstained section imaged using epifluorescence microscopy to visualize the results of fluorescent tracer injections. Three types of fluorescent neural tracers were injected into the brain to reveal the mesoscale neural connectivity. The four sets of sections: Nissl, myelin, CTB and fluorescent sections were processed and imaged separately, and later re-assembled computationally. A computational pipeline was established to perform high-throughput image processing. A common reference atlas (*Hashikawa et al., 2015*; *Paxinos et al.,*

*2012*) was registered to each individually reconstructed brain series and the projection strengths were suitably quantified.

## Experimental pipeline

All experimental procedures were approved by the Institutional Animal Care and Use Committee at RIKEN and a field work license from Monash University, and conducted in accordance with the Guidelines for Conducting Animal Experiments at RIKEN Center for Brain Science and the Australian Code of Practice for the Care and Use of Animals for Scientific Purposes. Female marmosets (*Callithrix jacchus*), 4 to 8 years old, 330 g - 440 g in weight, were acquired from the Japanese Central Institute for Experimental Animals.

### In-vivo MRI

Upon habituation, the marmosets promptly went through magnetic resonance (MR) imaging. MR scans were performed using a 9.4T BioSpec 94/30 US/R MRI scanner (Bruker, Biospin, Ettlingen, Germany) with actively shielded gradients that had a maximum strength of 660 mT/m. Several MRI protocols were carried out for each individual marmoset. T1 mapping and T2-weighted images (T2WI) were used in in-vivo MR imaging. More details of the scan protocol can be found in Appendix 1.

### Neuronal tracer injections

To conserve animals, four tracers were placed in the right hemisphere of each marmoset, including two anterograde tracers: AAV-TRE3-tdTomato (AAV-tdTOM) and AAV-TRE3-Clover (AAV-GFP), and two retrograde tracers: Fast Blue (FB) and CTB. Surgical procedures for tracer injections were adapted from the previously established protocols (*Reser et al., 2009*; *Reser et al., 2013*; *Reser et al., 2017*). Tracers were delivered at the injection sites using Nanoject II (Drummond, USA), with dosage controlled by Micro4 (WPI, USA). For cortical injections, each tracer was delivered with depths of 1200 µm, 800 µm, and 400 µm sequentially perpendicular to the cortical sheet, with equal volumes. The planning for tracer injections approximately followed a uniform 2×2×2 mm grid spacing, intended to cover the entire brain cortical and subcortical regions (*Grange and Mitra, 2011*) (Appendix 2). The current data set used to validate the method presented here includes 118 injections. At each injection site, one retrograde and one anterograde tracer was injected separately to cover the efferent and afferent projections of that site. *Figure 2a,b* shows currently covered injection sites.

### Ex-vivo MRI and cryo-sectioning

After tracer injection and a 4 week incubation period, the marmoset brain was perfused with a 0.1M phosphate buffer (PB) flush solution followed by 4% paraformaldehyde (PFA) in 0.1M PB fixation solution. The same MR scan protocol for in-vivo MRI was used for ex-vivo Diffusion Tensor Imaging (DTI) scanning. Additional high-resolution (300 µm) T2-weighted images (T2WI) were carried out for ex-vivo MR imaging (Appendix 1). Following fixation, the brain was transferred to 0.1M PB to take an ex-vivo MRI. It was then immersed in 10% then 30% sucrose solution over a 48 hr period to safeguard against thermal damage. The brain was embedded in freezing agent (Neg-50, Thermo Scientific 6505 Richard-Allan Scientific) using a custom developed apparatus and a negative cast mold of the brain profile. The apparatus was submerged in an optimal cutting temperature compound to expedite the freezing process (*Pinskiy et al., 2013*). More details can be found in Appendix 3.

Cryo-sectioning of the brain was performed using a Leica CM3050 S Cryostat in a humidity chamber set at 18°C and 80% humidity. The cryostat specimen temperature was set to −15 to −17°C while the chamber temperature was set to −24°C. This temperature differential was used to make certain the tissue was never in danger of being heated unnecessarily. Brains were cryo-sectioned coronally on a custom made cryostat stage using the tape transfer and UV exposure method (*Pinskiy et al., 2015*) (Appendix 4). Every four consecutive sections were separately transferred to four adjacent slides, to establish the four series of brain sections to be stained for different methods. Each section was 20 µm in thickness, hence the spacing between every two consecutive sections in the same series was 80 µm. The four slides were transferred and cured for 12 seconds(s) in a UV-LED station within the cryostat. All cured slides were placed inside a 4°C refrigerator for 24 hrs to allow thermal equilibrium.

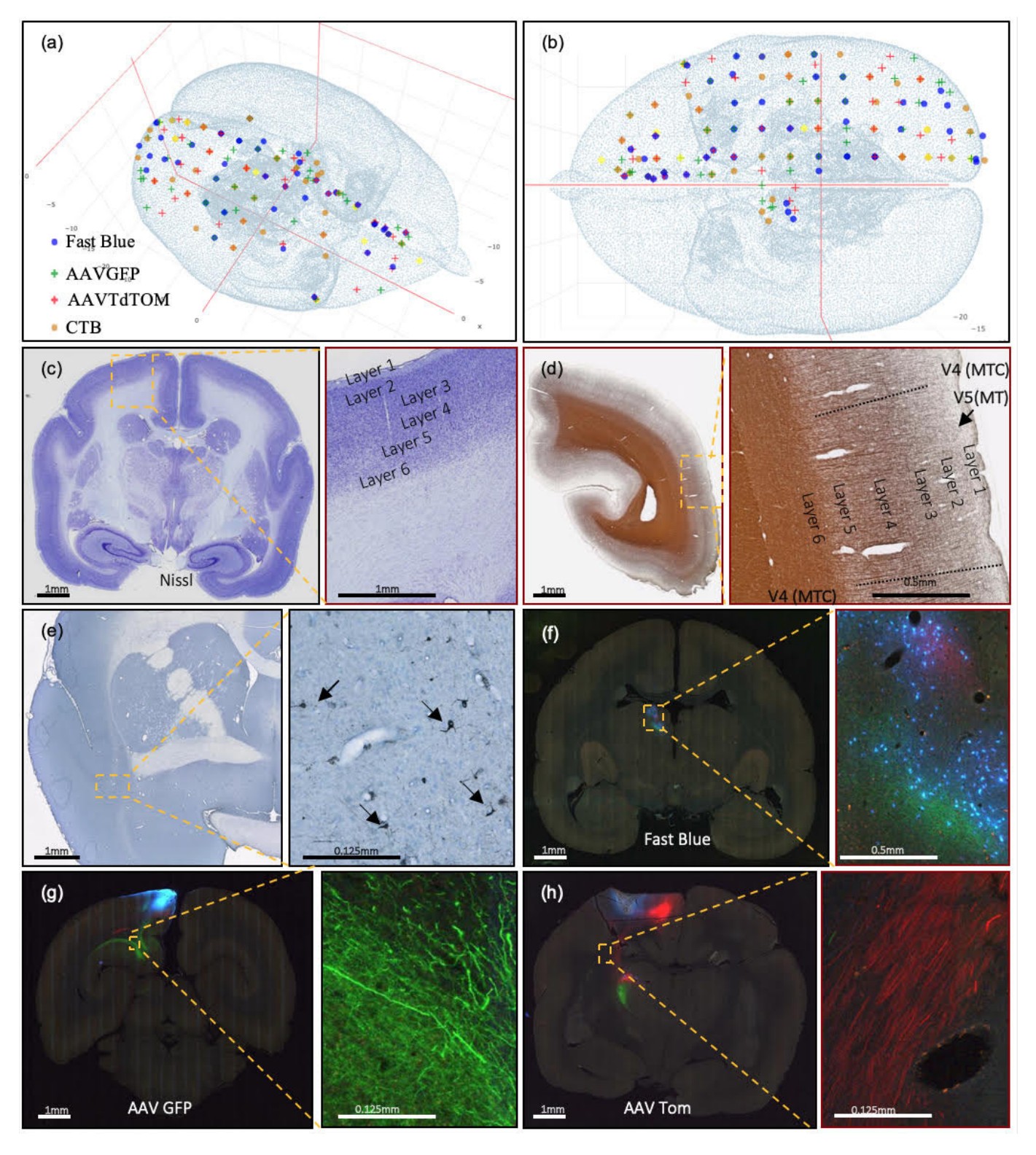

**Figure 2.** Current injection sites covered, example of staining methods, and different colors of marker in coronal brain sections. (**a, b**) Current successful injection sites using 2×2×2 mm grid spacing in the marmoset cortex in (**a**) 3D and (**b**) 2D dorsal view, in stereotaxic coordinates (*Paxinos et al., 2012*). (**b**) Current successful injection sites. Each tracer is represented with a different color of marker: blue: Fast Blue; green: AAV-GFP; red: AAV-tdTOM; brown: CTB. Two tracers, one anterograde and one retrograde, are injected at each site. (**c–h**) Sample coronal brain section images of four series. (**c**) A

*Figure 2 continued on next page*

*Figure 2 continued*

coronal section after Nissl staining is shown with increasing magnification. Around Area 3a (magnification box), six cortical layers and the white matter are clearly differentiable based on cell body density. (d) A coronal section of the left hemisphere after silver staining showing myelin. Around Visual area V4T (Middle Temporal) crescent; magnification box), layers I-VI can be clearly characterized based on the myelin fiber density. Heavy myelination can be seen in layer three and continues into layer 4–6 with clear inner and outer bands of Baillarger. (e) Partial coronal section after immunohistochemistry treatment for CTB. After injection into Area 10, CTB labeled neurons were found in the claustrum (magnification box). The arrows indicate CTB- labeled cells at 0.125 mm. (f–h) Coronal sections in different parts of the brain showing fluorescent tracers including (f) retrograde tracer Fast Blue (g) anterograde tracer AAV-GFP, and (h) anterograde tracer AAV-tdTOM.

DOI: https://doi.org/10.7554/eLife.40042.005

The following figure supplement is available for figure 2:

**Figure supplement 1.** Example of a coronal section of the brain showing fluorescent tracers in high magnification.

DOI: https://doi.org/10.7554/eLife.40042.006

## Histological staining

Separate histological staining processes were performed on the different series of brain sections (Appendix 5). High-throughput Nissl staining of neuron somata was performed in an automated staining machine (Sakura Tissue-Tek Prisma, DRS-Prisma-J0S) (*Figure 2c*). The myelin staining technique used a modified ammoniacal silver stain originally developed by Gallyas (*Gallyas, 1979*). The present modification provided higher resolution of fiber details that could be used for myeloarchitecture identification. A representative magnified image of myelin staining in the V4 (middle temporal crescent) visual cortex is shown in *Figure 2d*. Using a modified protocol developed for the MBA project at CSHL, the staining of retrograde and anterograde CTB labels were successfully attained (*Britto, 2000*) (*Figure 2e*). Finally, retrograde fluorescent tracers revealed originating somata while the anterograde tracers revealed projecting axons from fluorescent imaging. *Figure 2(f–h)* shows simultaneous fluorescent tract tracing using AAV-GFP, AAV-tdTOM and FB within the same brain. More detailed high-magnification images can be found in *Figure 2—figure supplement 1*.

The pipeline adopted the Sakura Tissue-Tek Prisma system for high-throughput staining purposes. Upon completion of auto staining, the system loaded the dehydrated slides into an automatic coverslipper (Sakura Tissue-Tek Glas, GLAS-g2-S0) where 24 × 60 mm cover glass (Matsunami, CP24601) were applied with DPX mounting media (Sigma, 06522); then put into drying racks for 24 hrs. *Figure 3* shows the overall steps as well as time taken to process one marmoset brain before moving to the computational pipeline starting with imaging.

Including imaging, one full Nissl brain series can be completed in 6 days. The myelin series including imaging requires 6.4 days using a limited 60-slide staining rack. The CTB series took a total of 7.9 days to complete due to batch limitations (3.5 batches with 120 slides/batch in total). The time for completion for the fluorescent brain series was 8 days; the slide scanning time on the Nanozoomer used in the project is approximately twice the brightfield scanning time. Overall, the four separate series of one brain could completed in two weeks (a pipeline processing rate can be found in Appendix 8). The digitized brains are then passed onto the computational pipeline including atlas registration, cell and process detection and online presentation.

## Computational pipeline

All the prepared slides were scanned by series with a Nanozoomer 2.0 HT (Hamamatsu, Japan) using a 20x objective (0.46 µm/pixel in plane) at 12-bit depth and saved in an uncompressed RAW format. Nissl, myelin and CTB series were brightfield scanned. Fluorescence series were scanned using a tri-pass filter cube (FITC/TX-RED/DAPI) to acquire the 3 RGB color channels for each slide. A Lumen Dynamics X-Cite *exacte* light source was used to produce the excitation fluorescence.

The RAW images for all four series of slides comprise ~8 terabytes of data for each brain. In order to process these large data volumes, the pipeline includes networked workstations for data-acquisition, image processing and web presentations. All systems were connected to two directly attached data storage nodes to ensure that all data were continuously saved and backed up. All components were integrated with 10 Gigabit Ethernet (10G network) to provide a cohesive solution (Appendix 6). The average node-to-node transfer rate was on the order of 250–450 MB/s, including limitations of hard disk speed.

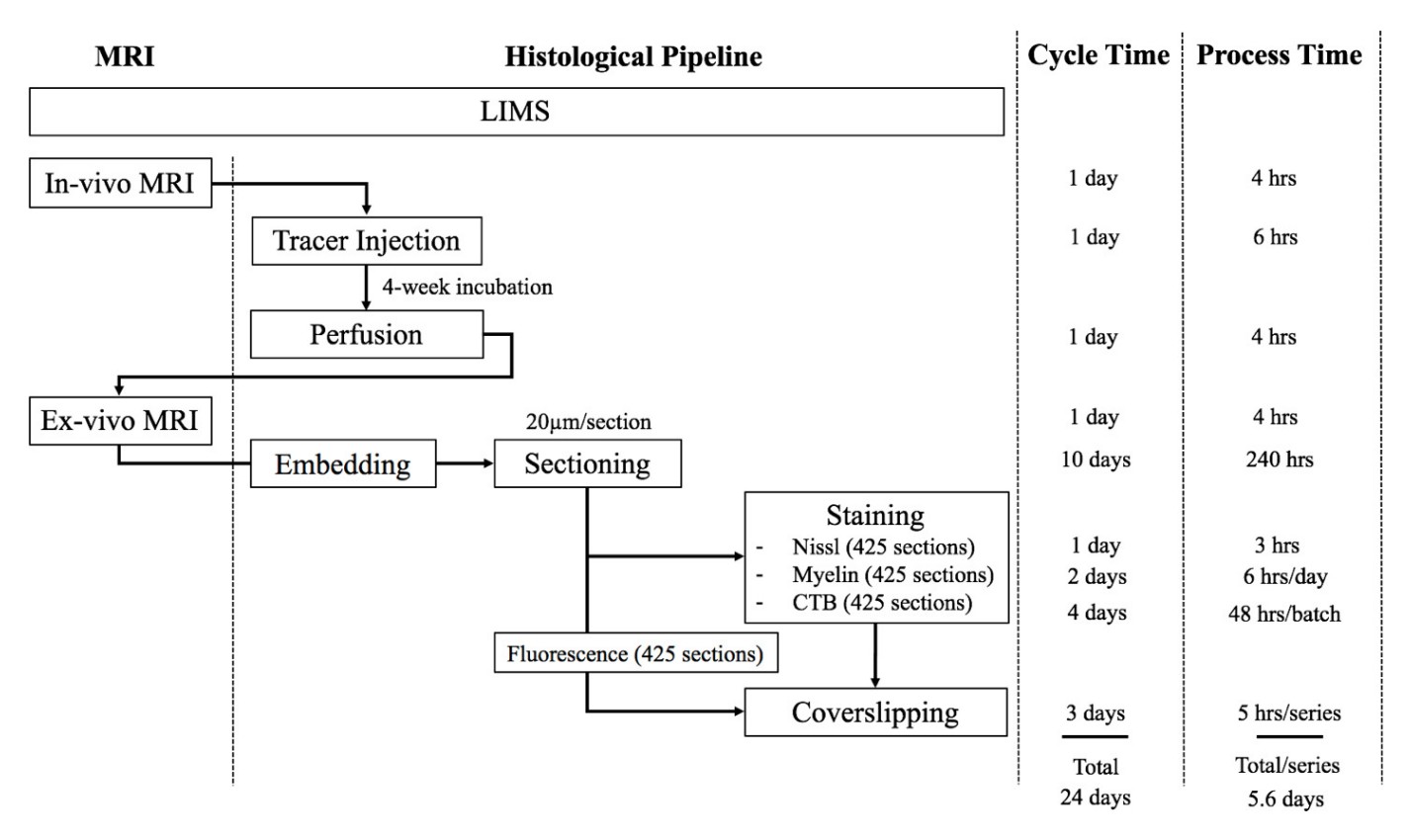

**Figure 3.** The workflow of the experimental pipeline and the processing time for one marmoset brain. Arrows show the sequence of individual experiments. A custom-made LIMS (Laboratory Information Management System) performs housekeeping for the entire process and constitutes an electronic laboratory notebook. The entire brain is sectioned into ~1700 sections,~400 in each series. Each series include ~295 slides, comprising of 1/3 of the slides with two brain sections mounted and 2/3 with one brain section/slide. Coverslipping includes the drying and clearing stages. The processing time does not include the overnight waiting period after sectioning in each batch. The overnight incubation time is excluded in the CTB procedure as well as the overnight dehydration in a myelin stain. Process Time on the right shows the time involved in processing each experimental step, in hours. The Cycle Time (in days) shows the total time required to initiate and finish the entire procedure from start to finish, including quiescent periods, before commencing the procedure for another brain. Total time on the bottom is not a summation of the individual procedure times above because of parallel, pipelined processing which reduces total processing times. For example, when Nissl series are being processed in the automatic tissue staining machine for Nissls, CTB and myelin staining can be performed simultaneously at other workstations.
DOI: https://doi.org/10.7554/eLife.40042.007

Imaging data were collected from the Nanozoomer and then automatically transferred to a data acquisition system. This step ensured uninterrupted scanning regardless of the limited disk space on the Nanozoomer system relative to the amount of data being acquired. The data acquisition system is the central repository for image pre-processing including image cropping, conversion, and compression (Appendix 7).

The quality control (QC) service was applied to all stages of experimentation and image data flow in order to correct and improve the pipeline process organically. The experimental pipeline process information was recorded in an internal Laboratory Information Management System (LIMS). It supported the workflow by recording the detailed status of each experimental stage for each brain. Similarly, a separate online QC portal dictated all the image pre-processing stages (*Figure 4*). Through the LIMS and QC portal, it was possible to flag damaged sections to avoid unnecessary post-processing and identified the need to repeat a specific processing stage.

Image registration, cross-modal registration and automatic annotation, and tracing signal detection were performed in the image processing server. Images of individual sections were down-sampled by 64 times and registered to one another using rigid-body transformation (*William et al., 2011*). Registered 2D images were used to create a 3D volume of the brain in NIfTI format (NIfTI-1

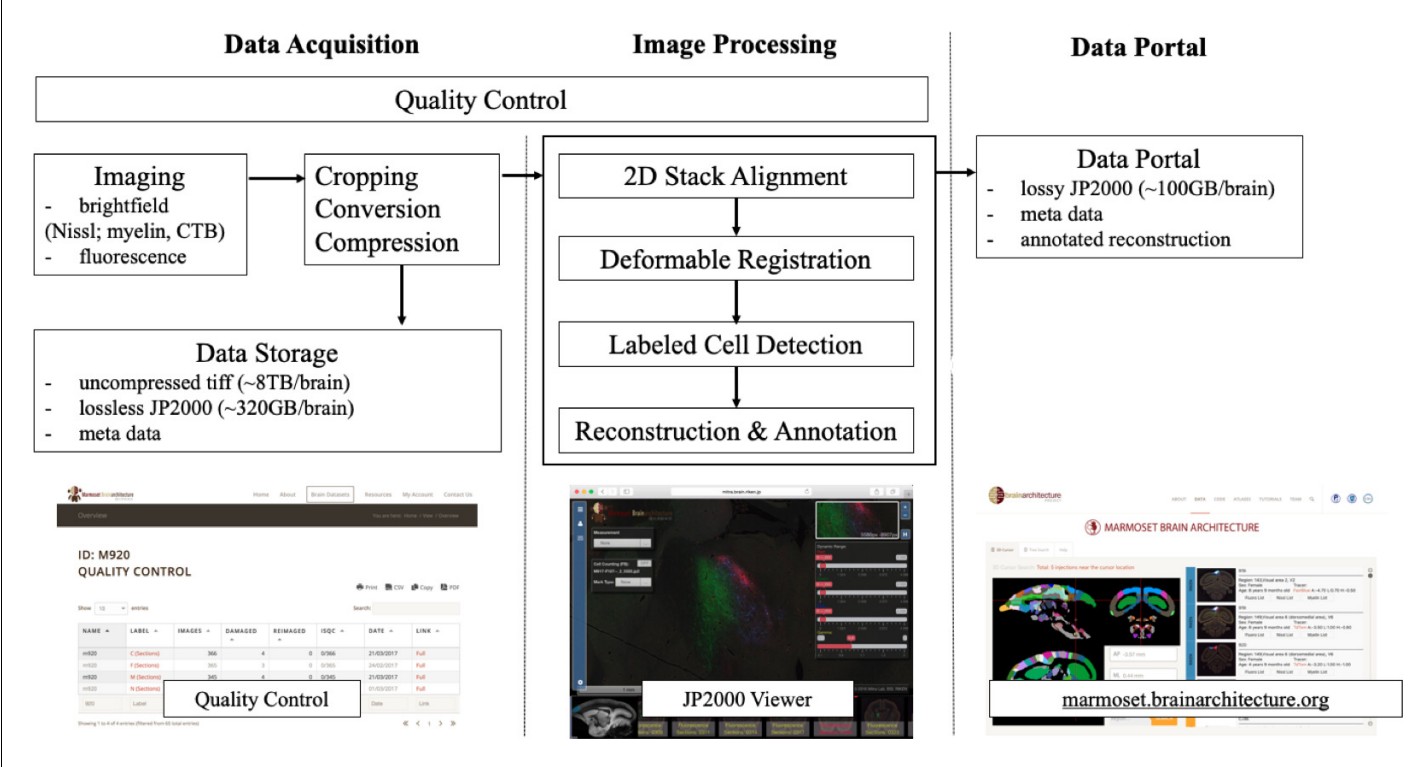

**Figure 4.** A flow chart showing the workflow of the computational pipeline, from data acquisition to image processing and finally dissemination on the public data portal. Arrows show the data flow. A quality control system is implemented at every stage of the pipeline until final data release. The display of the data portals is to allow interactive service. (a) A quality control site (snapshot on the bottom left) which helps improve the pipelines process speed and manually flags unnecessary sections to avoid further post-processing. (b) An Openlayer 3.0 JPEG2000 viewer (snapshot on the bottom middle) including several controls such as dynamic range, gamma, measurement and auto cell detection tool to allow for a users' interpretation (*Lin et al., 2013*). (c) The data portal site (snapshots on the bottom right) helps to host all successful and processed dataset for publishing purposes.
DOI: https://doi.org/10.7554/eLife.40042.008

Data Format, 2016) for each series. The transformation matrix for each downsampled image was applied to the corresponding full resolution image.

The brain outline of Brain/MINDs atlas (*Woodward et al., 2018*) was applied to the downsampled images after 2D registration to separate the brain regions from background and ventricles. Automatic annotation of the brain structures was achieved by registering the Brain/MINDs atlas to ex-vivo MRI and then aligned to the 2D registered Nissl series ('target images'). A 3D global affine transformation was applied to move the atlas images into the coordinate space of the MRI images. After transformation, the atlas images was matched to the MRI images using Large Deformation Diffeomorphic Metric Mapping (LDDMM) (*Ceritoglu et al., 2010*) which transforms the atlas coordinate to the MRI image coordinate system. The same method was applied again to the transformed atlas images in order to match the target Nissl images. Individual brain regions could be automatically identified based on the transformed atlas. *Figure 5a* shows the example of automatic registration from Brain/MINDs atlas to target Nissl images. Cross-series registration using Euler2DTransform from Insight Segmentation and Registration Toolkit (*ITK, 2017*) was performed to align 64-time downsampled myelin, CTB and fluorescence series of images to target Nissl images (*Figure 5b–d*). Finally, the transformation matrices calculated from the downsampled images were applied to the corresponding full resolution images. The annotations from the transformed atlas were aligned with the histology images of each series.

Injection volume was estimated by measuring the tracer spread at the injection site. Automatic cell and process detection was applied to individual registered sections in order to compute a draft whole-brain connectivity matrix. As an integral part of the computational pipeline, a data portal was developed to allow for viewing and interpreting high-resolution images online (http://marmoset.

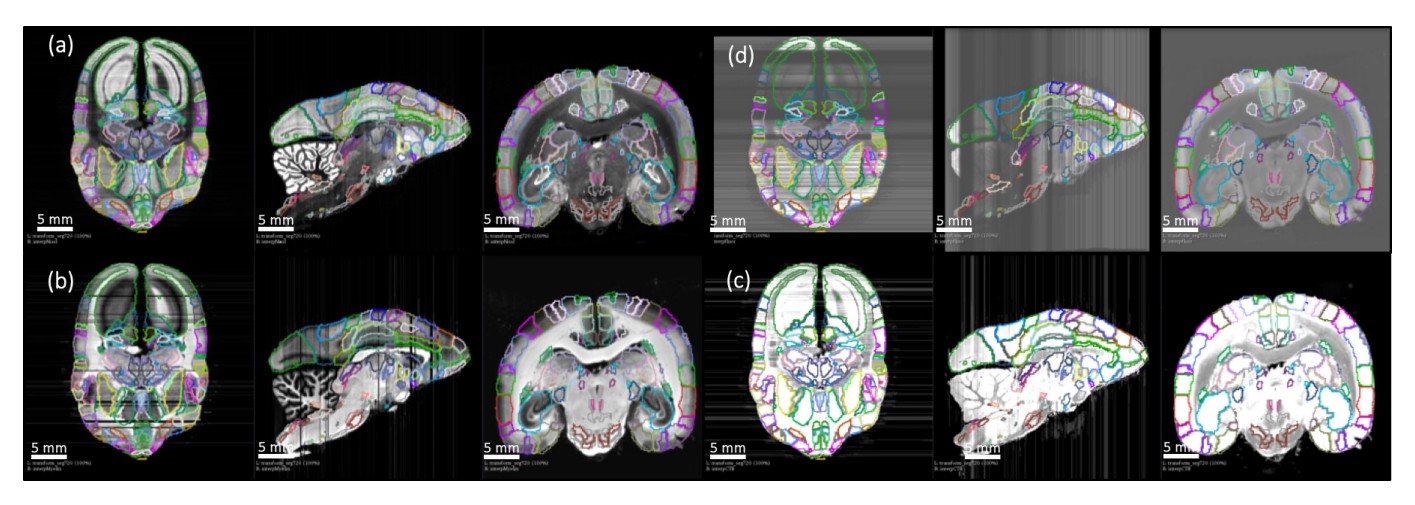

**Figure 5.** 3D deformable registration and atlas mapping of all four series. The Brain/MINDs atlas was registered with ex-vivo MRI volume, and subsequently registered to target Nissl series (**a**) The shaded areas indicate missing sections at the end of processing (quality control). Other series including (**b**) myelin, (**c**) CTB and (**d**) fluorescence series were cross-registered to target Nissl series, and aligned with the atlas annotations. Only gray scale images are shown and they are sufficient for the registration process. Sample sections in transverse (left), sagittal (middle), and coronal (right) were shown for each series.

DOI: https://doi.org/10.7554/eLife.40042.009

brainarchitecture.org). By incorporating an Openlayer 3.0 image server with a custom image viewer, the data portal allows fully interactive zoom and pan, supports online adjustment of RGB dynamic range and contrast, as well as gamma adjustment (*Figure 4*). The data portal also provides visualization of cell detection results and an interactive tool for injection volume measurement.

## Successful re-assembly of 3D volumes

In order to evaluate the quality of the image registration pipeline, we applied computational approaches to separately register series acquired for individual data modalities into separate volumes. Both high-quality and low-quality section images with staining issues, image variation, or artefacts were considered in the process. Adoption of the tape transfer method allowed us to maintain the geometry of the brain sections in the high-quality 20 µm section images. This allowed successful section-to-section (2D) alignment using only rigid-body transformations. Poor-quality sections such as sections with folding, tears, artefacts and discoloration missed from the previous QC stage were selected by visual inspection and excluded from the 2D alignment step. Less than one percent of total sections were excluded. *Figure 6* (left) shows one marmoset brain with different staining procedures in coronal, sagittal and transverse planes after image reconstruction. It also shows the results of how the geometry of the brain has been maintained in each series.

## Atlas registration

Using external references such as the same-subject ex-vivo MRI or the population-typical reference atlas (*Woodward et al., 2018*), we aimed to reconstruct the true shape of the subject brain and to avoid the classical curvature recoverability problem of sectioned objects. This atlas-informed reconstruction (*Lee et al., 2018*) improved reconstruction accuracy compared to the atlas-uninformed neighbor-to-neighbor method, as well as reduced the deformable metric cost. The impact of the ex-vivo MRI constraint on the 3D reconstruction is shown in *Figure 6* (right). A visible distortion is present in the MRI-unguided reconstruction. The degree of shrinkage is 7% from *in-* to ex-vivo MRI and 1% from ex-vivo MRI to histology. This distortion is corrected by a MRI-guided method using a reference atlas. The MRI-constrained alignment of the Nissl sections produces a Nissl volume which closely resembles the convex hull of the same-subject MRI, leading to accurate parcellation of the brains in question.

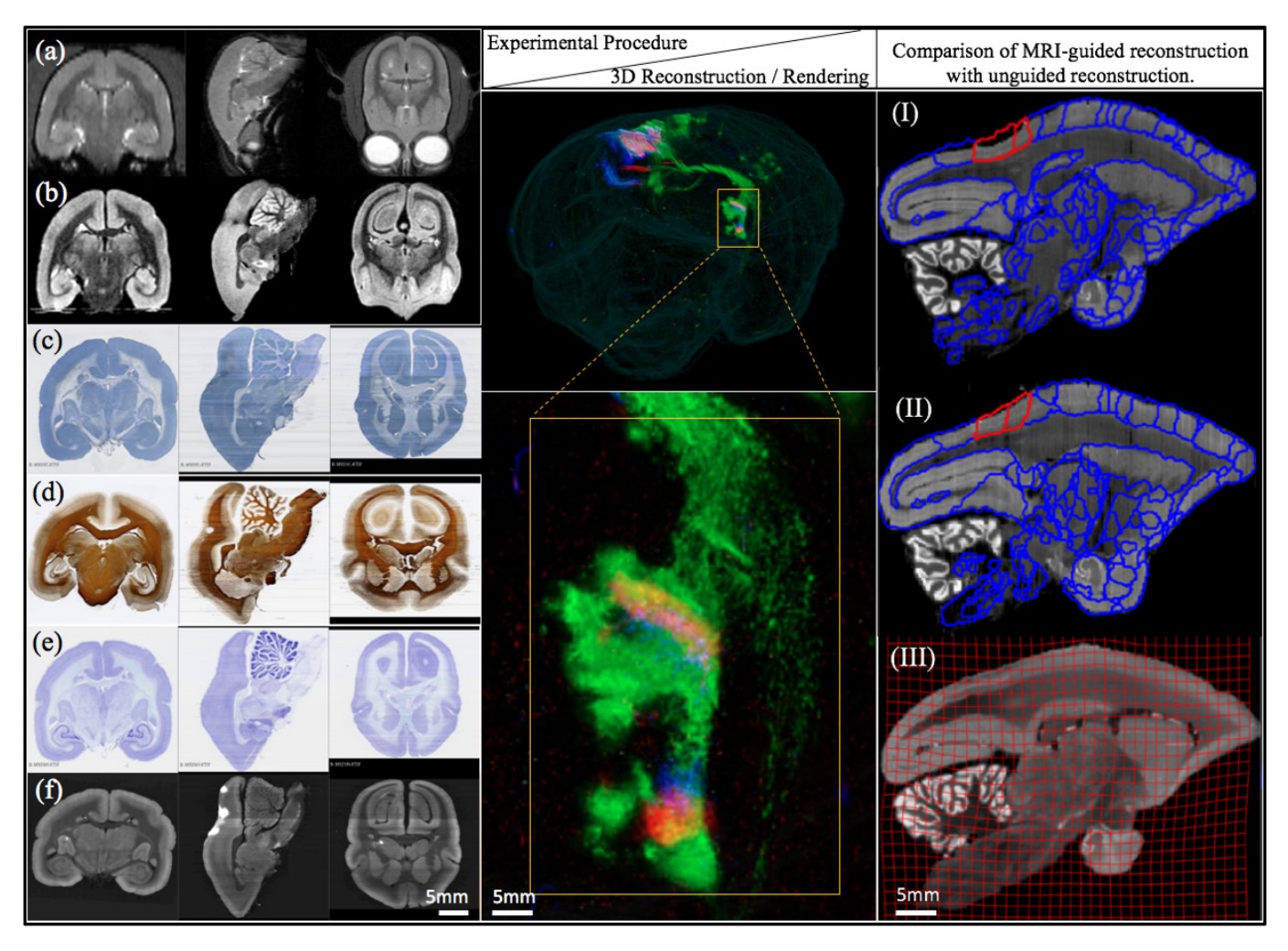

**Figure 6.** Different stages of image acquisition, 3D reconstruction, and MRI-guided registration in this experimental protocol. (left) Views of one marmoset brain after each experimental protocol. (a) in-vivo MRI (b) ex-vivo MRI (c) CTB staining (d) myelin staining (e) Nissl staining (f) fluorescence imaging. Coronal, sagittal and transverse planes at the same (MRI) or consecutive sections (staining series) are shown with 3D registration and reconstruction. (middle) A 3D visualization of the fluorescent tracer projection. Simultaneous anterograde (red, green) and retrograde (blue) tracing reveals a reciprocal connection between the dorsal medial visual area (injection site) and the thalamus (anterograde projection and retrograde cell labeled sites) especially lateral posterior nucleus and lateral pulvinar. The connectivity can be observed with this 3D visualization which shows the pathway of tracers in through the brain volume. (right) Comparison of MRI-guided reconstruction with unguided reconstruction. I: the target Nissl stack reconstruction by unguided piecewise neighbor-to-neighbor alignment. II: the MRI-guided reconstruction. III: same- subject T2-weighted MRI.
DOI: https://doi.org/10.7554/eLife.40042.010

## Results

Brain volumes generated by the combined pipeline were further subjected to automated cross-modal registration and atlas segmentation, to obtain a regional connectivity matrix.

### Connectivity mapping

The registration process permitted brain surface reconstruction (*Video 1*), 3D visualizations of projections, and virtual cuts in other planes of section than the original coronal sections (*Figure 6*; right). After segmentation and registration, we derived quantitative values of tracer signals within each region. We developed an image processing method for detecting axonal and dendritic fragments in images, and applied it to each high resolution section (0.46 µm) to segment the anterograde projections. The segmented pixels were appropriately weighted to create an isotropic 3D summary of the

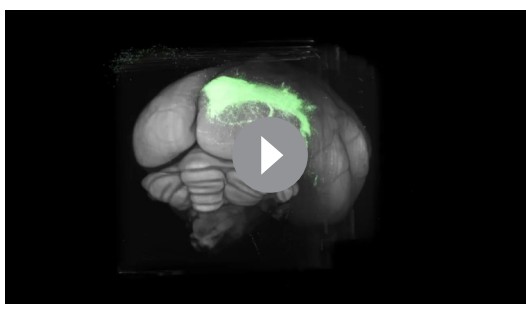

**Video 1.** The registration process permitted brain surface reconstruction. A brain fully reconstructed using MRI guided registration with process and cell detection. A clear pathway is seen from the tracer traveling from region to region in this 3d visualization of projections. Virtual cuts in planes of sections other than the original coronal sections are also shown.
DOI: https://doi.org/10.7554/eLife.40042.011

projections (*Markov et al., 2014*). We developed an automatic cell detection method (*Pahariya et al., 2018*) to segment somata labeled by the retrograde label Fast Blue throughout the entire brain. Injection sites were separated out from the rest of the brain. The projection strength between each target and source region was quantified as the fractional number of voxels containing tracer labels.

The registration process together with process and cell detection methods allowed us to obtain intermediate resolution, annotated images for each tracer and to review the atlas parcellation. *Figure 7* shows the result of three fluorescent tracer injections in the same animal and their origin/projections, resulting in one column and two rows in the putative connectivity matrix. In this example, Fast Blue, AAV-GFP and AAV-TdTOM were injected in V6, V1, and V6 visual cortex respectively. Automatic process detection identified projection targets from V1 to various regions, including the most prominent projections detected in V5 and dorsal lateral geniculate

**Figure 7.** A part of the connectivity matrix identified with tracer injections in one sample brain. The retrograde tracer Fast Blue was injected in V6 and found in high density in several regions such as lPul and A31. AAV-GFP was injected in V1 and AAV-TdTOM in V6 and show clear projections to the thalamus and other visual areas. Each row contains all projections to different brain regions originating from those AAV tracers. The magnified images highlight some clear origin/projections from the injected tracers in the connectivity matrix.
DOI: https://doi.org/10.7554/eLife.40042.012

nucleus (DLG). Projection targets from V6 included the lateral pulvinar (LPul) and medial pulvinar (MPul) among other targets. Automatic cell detection for the Fast Blue tracer identified the regions projecting to V6 including prominent projections from A6DC, A31, and inferior pulvinar (IPul).

## Discussion

We have described a high throughput, standardized pipeline integrating experimental and computational elements into a unified system and workflow for processing tracer-injected marmoset brains, representing an essential step towards producing a whole-brain mesoscale connectivity map in an NHP. The pipeline combines the well-established neuroanatomical protocols with automated instrumentation and a software system for greatly improving the efficiency of the techniques compared to conventional manually-intensive processing. Access to high-quality in-vivo and the ex-vivo MRI provided us with important auxiliary data sets facilitating re-assembly of the section images and atlas mapping, thus ameliorating the challenges arising from increased individual variations in brain geometry in an NHP compared with laboratory mice.

It is important to compare with other microscopic methods that have become established in recent years for light-microscope based anatomy, including serial block-face two photon scanning microscopy (*Denk and Horstmann, 2004*; *Osten and Margrie, 2013*; *Svoboda and Yasuda, 2006*) and light sheet microscopy (*Glaser et al., 2018*; *Nikon, 2018*), as well as knife-edge scanning microscopy (*Mayerich et al., 2008*). While these methods have important advantages, particularly the reduced need for section-to-section registration to produce the initial 3D volumes for further analysis, the classical methods have the important advantage of carrying through conventional histo-chemistry without major protocol alterations, producing long-lasting stains and precipitates that can be imaged using brightfield microscopy. Classical Nissl and myelin stains remain the gold standard for cytoarchitectonic texture-based determination of precise brain region location and delineation. These series are produced routinely with ease in the pipeline. The thin physical sections can be imaged rapidly in whole-slide imaging scanners and at relatively high numerical aperture (resolution in light sheet microscopy is comparatively limited due to reduced NA in the bulk of the sample).

### Individual variation in brain anatomy in the marmoset

Previous studies aimed at generating population based atlases on non-human primates (*Black et al., 2001b*; *Black et al., 2001b*; *Feng et al., 2017*; *Hikishima et al., 2011*; *Quallo et al., 2010*) have focused on mapping individual brains to a common mean template. Individual variations were addressed in terms of variation in stereotaxic coordinates of major landmarks such as sulci (*Black et al., 2001a*; *Black et al., 2001b*; *Hikishima et al., 2011*). A few studies have explicitly reported variations in brain sizes (*Hikishima et al., 2011*) but we did not find an analysis of variations of individual regions, or co-variations across regions.

The data gathered in the project permits an a-posteriori analysis of individual variations in brain anatomy and cytoarchitecture. While a comprehensive analysis has to be left for a future study using this data, we summarize a few observations based on a partial analysis. Within a sample of 26 cases, the whole brain volume had a median of 8222.5 $mm^3$ with a median absolute deviation (MAD) of 319.4 $mm^3$. In comparison to the Paxinos/Hasikawa (Brain/MINDS) template (*Hashikawa et al., 2015*; *Hikishima et al., 2011*; *Woodward et al., 2018*), our animals were older and mostly heavier than the template brain animal. Yet the brain sizes were similar to the template brain. We did not find a significant relationship between whole brain volume and age or body weight (see Appendix 10) within our data set. Nevertheless, some individual compartment sizes significantly departed from the template brain (e.g. the Hippocampal formation showed a consistently smaller size), indicating that the template brain may not be representative of a population average. Quantitative analysis of the covariation of cytoarchitectonic structure across the whole marmoset brain, in a significantly sized sample, is possible with the data gathered in the current study and will be carried out in the near future. We expect that the reference atlas may need to be revised based on the results of such a study.

### Injection size variations and localization within compartments

Based on a preliminary analysis, 73% of the injections placed are localized within atlas-determined anatomical compartments, whereas 27% showed some spread across boundaries. Manual analysis of

a subset of 15 injections showed diameters in the range 0.8 mm-2.5mm, indicating rough correspondence with the desired grid spacing. Among these 15 injections, six had tracer spread beyond the compartment boundary. On average, for these six injections, about 68% of the volume was restrained within the same region as the injection center, while about 32% of the volume leaked outside to adjacent regions.

## Combining injections with those from previous studies to increase sample size

We were able to combine subsets of the injections placed in this study with injections in previous studies, as well as data gathered in collaborating laboratories, to generate and test specific hypotheses, indicating the utility of the data gathered in the project (*Lee et al., 2018*; *Majka et al., 2018*). In addition, analysis of injection centers show proximity/overlap of injections from a previous data set from the Rosa laboratory for which 3D spatial information is available (Appendix 10). This should permit virtually increasing N for this project.

## Completion of Brain-wide coverage in the marmoset

An estimate of the total number of injections that will provide brain-wide coverage, in the hybrid grid-approach adopted in the paper depends on the lower cutoff placed on atlas compartments to be injected. To obtain an upper bound, we assume a cutoff of 0.8 mm³ (corresponding to the smallest injections we placed so far), which corresponds to 356 sites (712 injections). So far, 190 injections have been placed in 49 brains. To cover the rest of the brain, 264 more injections would be placed in the cortex, and 258 injections in subcortical regions and cerebellum. This would require 131 brains. The current pipeline has achieved a maximum capacity of 2 brains/month. At this rate, a complete marmoset mesoscale connectivity map would be available by 2024. However, we expect that the process can be sped up considerably by multiple groups working together in a collaborative manner using similar methods. Such a project would necessarily need to have international scope and can be expected to be transformative for our understanding of primate brain architecture.

## Larger brains

The pipeline described here is for $1 \times 3$ inch glass slides that fortunately are large enough to accommodate marmoset brains in coronal section. The pipeline can be generalized in the future to $2 \times 3$ inch slides, which can handle larger brains (such as that of macaque), with a few technical innovations, importantly in stainers/coverslippers for the larger format slides. This should allow the easy and economical neurohistological processing of larger sized vertebrate brains, opening up the possibilities of applying modern computational neuroanatomical techniques to a significantly broader taxonomic range of species, allowing for the study of comparative neuroanatomical questions with unprecedented computational depth.

## Acknowledgements

The authors would like to acknowledge Tetsuo Yamamori, Akiya Watakabe and Hiroaki Mizukami at RIKEN Center for Brain Science (CBS) for providing the virus tracers. We thank Noritaka Ichinohe and Yoko Yamaguchi at RIKEN CBS as well as Daniel Ferrante from CSHL Mitra Lab for helping with computational aspects of the pipeline. We acknowledge the effort from our previous technicians, Kevin Weber and Khurshida Hossain (RIKEN) and the support from Erika Sasaki at the Central Institute for Experimental Animals for providing marmosets used in this project. We thank staff of the Biomedicine Discovery Institute, (Monash University) for providing a detailed protocol for injection surgeries (Katrina Worthy), helping perform surgeries (Jonathan Chan), and providing helpful insight on the development of the web infrastructure (Shi Bai and Piotr Majka). We acknowledge James Bourne, Inaki-Carril Mundinano and William Kwan (Australian Regenerative Medicine Institute, Monash University), who performed MRI-guided stereotaxic brain injections in deep brain structures. We acknowledge the support from Jaikishan Jayakumar at IIT Madras on neuroanatomical questions, as well as Xu Li from CSHL machine vision algorithms for process detection. This project is supported by the Brain Mapping of Integrated Neurotechnologies for Disease Studies (Brain/MINDS) from the Japan Agency for Medical Research and Development, AMED under grant number JP17dm0207001, the Crick-Clay Professorship (CSHL), the Mathers Foundation, the H N Mahabala

Chair at IIT Madras, the ARC Centre of Excellence for Integrative Brain Function and Monash University.

The authors would also like to thank the reviewers for constructive input which significantly improved the manuscript from its original version.

## Additional information

### Funding

| Funder | Grant reference number | Author |
|---|---|---|
| Australian Research Council | CE140100007 | Marcello GP Rosa |
| Japan Agency for Medical Research and Development | JP17dm0207001 | Hideyuki Okano |
| Clay Mathematics Institute | Crick-Clay Professorship | Partha Mitra |
| Indian Institute of Technology Madras | HN Mahabala Chair | Partha Mitra |
| Mathers Foundation | | Partha Mitra |

The funders had no role in study design, data collection and interpretation, or the decision to submit the work for publication.

### Author contributions

Meng Kuan Lin, Conceptualization, Data curation, Formal analysis, Methodology, Project administration, Writing—original draft, Writing—review and editing; Yeonsook Shin Takahashi, Jaimi Nagashima, Methodology, Writing—review and editing; Bing-Xing Huo, Methodology, Formal analysis, Writing—review and editing; Mitsutoshi Hanada, Junichi Hata, Keerthi Ram, Michael I Miller, Methodology; Alexander S Tolpygo, Methodology, Project administration; Brian C Lee, Methodology, Formal analysis; Marcello GP Rosa, Methodology, Writing-review and editing; Erika Sasaki, Resources; Atsushi Iriki, Writing-review and editing; Hideyuki Okano, Supervision, Project administration, Writing—review and editing; Partha Mitra, Conceptualization, Methodology, Supervision, Formal analysis, Writing—original draft, Writing—review and editing

### Author ORCIDs

Meng Kuan Lin (iD) http://orcid.org/0000-0002-8191-8563
Marcello GP Rosa (iD) http://orcid.org/0000-0002-6620-6285
Hideyuki Okano (iD) http://orcid.org/0000-0001-7482-5935
Partha Mitra (iD) http://orcid.org/0000-0001-8818-6804

### Ethics

Animal experimentation: The experiment protocol was approved by the Research Resource Division (RRD) under (approval authorization H29-2-242(3)) from the support unit for animal resources development in conformity with Article 24 of the RIKEN regulations for animal experiments in Center for Brain Science, RIKEN. Each marmoset received multiple injections of fluorescent tracers using stereotaxic coordinates. All brain surgery was performed under isoflurane (2%)/alfaxan (100ul/dose) anesthesia and every effort was made to minimize suffering. Body temperature, heart rate, and SPO2 were continually monitored during surgery.

### Decision letter and Author response

Decision letter https://doi.org/10.7554/eLife.40042.037
Author response https://doi.org/10.7554/eLife.40042.038

# Additional files

## Supplementary files

• Supplementary file 1. List of target structures and number of injections.
DOI: https://doi.org/10.7554/eLife.40042.013

• Supplementary file 2. Reference list of trace tracing studies.
DOI: https://doi.org/10.7554/eLife.40042.014

• Transparent reporting form
DOI: https://doi.org/10.7554/eLife.40042.015

## Data availability

All data generated through this pipeline is continually available from web portal: http://marmoset.brainarchitecture.org/.

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

# Appendix 1

DOI: https://doi.org/10.7554/eLife.40042.016

## MRI Method

During MRI, the animal was anaesthetized using 3% (±1%) isoflurane in oxygen and received an intraperitoneal injection of sterile saline (3 ml) to avoid dehydration during the procedure. Throughout the entire procedure, a mixture of oxygen and 2% (±0.5%) isoflurane was administered to maintain anesthesia. A custom-made head holder (Qualita Ltd., Saitama, Japan) was used to fix the marmoset head within the imaging tube, such that the rostral-caudal axis of the head was stereotaxically aligned with the tube. A small glass capillary with a contrast agent was used in each ear bar, such that the positions of the ear bars would be visible in the MRI images. A heating pad was used to maintain the body temperature. Heart rate, blood oxygen saturation levels, rectal temperature and respiration rate were continuously monitored and recorded every 10 min.

During **in-vivo MR** imaging, high-resolution 3D T1 mapping was carried out using a Magnetization-Prepared Rapid Gradient-Echo (MPRAGE) sequence (**Liu et al., 2011**) with a repetition time (TR) = 6000 ms, inversion times (TI) = 150, 1300, 4000 ms, (TE (echo time)=2 ms TD (time-domain) = 9 ms) and a nominal flip angle (FA) = 12 degrees. Imaging planes were axial slices with FOV = 48.0 × 38.4×22.6 mm at matrix size = 178 × 142×42. T2-weighted images (T2WI) were acquired using a rapid acquisition with relaxation enhancement sequence (**Hennig et al., 1986**) with the following parameters: repetition time/echo time echo = 4000 ms/22.0 ms, RARE factor = 4, averages = 3, field of view = 48 mm×48 mm, matrix = 178 × 178, slice thickness = 0.54 mm.

Diffusion weighted images were acquired by a pulse-field gradient spin-echo (PGSE, the Stejskal-Tanner diffusion preparation (Stejskal & Tanner, 1965)) based on echo planner imaging sequences along 30 MPG axes and were acquired with the following parameters: b-values = 1000 s/mm$^2$, TR = 4000 ms, TE = 25.57 ms, averages = 3, k-space segments = 6, matrix = 128×128, FOV = 44.8×44.8 mm$^2$, and slice thickness = 0.7 mm. The DTI map was acquired using a method adapted from Fujiyoshi et al. (**Fujiyoshi et al., 2016**). An eigenvector associated with the largest eigenvalue λ1 was assumed to represent the local fiber direction. Three DTI maps were reconstructed from the data as follows: axial diffusivity (AD) = $\lambda_1$, radial diffusivity (RD) = $(\lambda_2 + \lambda_3)/2$, and mean diffusivity (MD) = $(\lambda_1 + \lambda_2 + \lambda_3)/3$.

For the **ex-vivo MR** imaging following perfusion, the brain was immersed in an electronic liquid (Fluorinert FC-72; 3M) in a 32 mm ID acrylic tube. High resolution T2-weighted images (T2WI) were acquired using a rapid acquisition with relaxation enhancement sequence (**Hennig et al., 1986**) with the following parameters: repetition time/echo time echo = 10000 ms/29.36 ms, RARE factor = 4, averages = 16, field of view = 36 mm×30 mm, matrix = 360 × 300, slice thickness = 0.2 mm. Diffusion weighted images were acquired by a pulse-field gradient spin-echo (PGSE, the Stejskal-Tanner diffusion preparation (**Stejskal and Tanner, 1965**)) based echo planner imaging (**Mansfield and Pykett, 2011**) sequence along 128 MPG axes which was acquired with the following parameters: b-values = 1000, 3000 and 5000 s/mm$^2$, TR = 4000 ms, TE = 28.4 ms, averages = 2, k-space segments = 10, matrix = 190×190, FOV = 38.0×38.0 mm$^2$, and slice thickness = 0.2 mm.

## Appendix 2

DOI: https://doi.org/10.7554/eLife.40042.016

### Tracer Injections

Our project plans to cover 255 injection sites in the marmoset brain, one anterograde and one retrograde tracer at each site, evenly distributed across the grey matter of the right hemisphere of the marmoset brain. The stereotaxic coordinates of all injection sites were systematically chosen using an MRI-based atlas (*Hashikawa et al., 2015*) and the injection location choice was based on an established algorithm (*Mitra, 2014*). Briefly, the right hemisphere was separated into 255 equal sized parcels, respecting anatomical boundaries. The plan resulted in 199 injection sites within the cerebral cortex, and 56 injection sites in the subcortical regions. Each subcortical region was evaluated in terms of the structure's volume. The injection was then placed based on the grid space modeled for the individual structure of interest.

We used a borosilicate micropipette with an outer diameter of 20–30 µm as a vector of injection. The tracer was placed at each appropriate depth with an injection speed of 20 µl/min. Anterograde tracers, AAVTRE3TdTom (0.3 µl) and AAVTRE3Clover (0.3 µl) and retrograde tracers, Fast Blue (FB, 0.3 µl 5% solution in distilled water; Funakoshi; Tokyo, Japan) and biotin conjugated Cholera toxin subunit B (CTB, 0.6 µl 1% Enzolife, New York, USA) were used.

Post recovery, the animal was housed individually and monitored throughout the 4 week incubation period. The animal received a non-steroidal anti-inflammatory (Oral Metacam; 0.05 mg/kg, Boeringer Ingelheim) for three days immediately following the surgery.

# Appendix 3

DOI: https://doi.org/10.7554/eLife.40042.016

## Perfusion/Embedding

After the 4 week viral incubation period, the animal was euthanized and perfused. The marmoset was injected with diazepam (Pamlin:2 mg/kg), ketamine (10 mg/kg), then pentobarbital (80 mg/kg) to anesthetize. The animal was then perfused using an 18G oral gavage needle that entered the left ventricle and terminated at the aorta through the aortic valve. 500 mL of heparinised PBS was used (50 ml/min) to remove the blood supply prior to the beginning of asystole to ensure that no clotting occurred; afterwards 500 mL of 4% PFA in 0.1M PB was used (70 ml/min) for fixation purposes.

After extraction, the brain was submerged in 4% PFA overnight. The brain was then transferred to 0.1M PB and underwent a post-mortem ex-vivo MRI. Following the ex-vivo MRI, the brain was transferred into 10% sucrose in 0.1M PB overnight and then placed in 30% sucrose in 0.1M PB for means of temperature protection.

A rectangular base mold was custom made with Polylactic Acid (PLA) at $3 \times 4 \times 5$ mm. A slit was opened from the bottom of the mold and an additional piece of PLA was cut to fit into the slit for easy removal of the brain block after the freezing process. A 3D-printed brain mold made from MR images of several marmoset brains (*Hashikawa et al., 2015*) was attached to a positioning bar with its rostral side facing the arrow direction (*Appendix 3—figure 1*). A custom freezing platform, also 3D-printed, secured the base mold flat and allowed the positioning bar for the brain mold (dorsal side down) to adjust vertically.

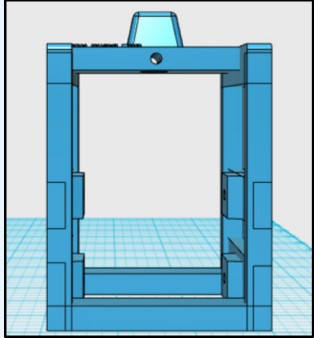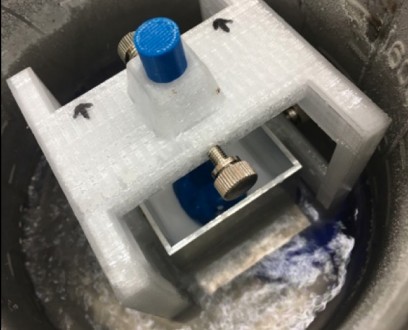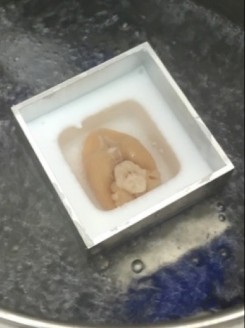

**Appendix 3—figure 1.** Rectangular base mold was designed and printed to serve as a freezing platform. The freezing platform was used to control the position of the brain mold to the base mold during freezing. The positioning bar is adjustable to allow ease of insertion and removal of the brain mold from the base mold.

DOI: https://doi.org/10.7554/eLife.40042.019

While the brain and base molds were attached to the freezing platform, the positioning bar was adjusted to lower the brain mold dorsal (down) side to 2 mm from the slit. Embedding medium Neg50 (Richard Allen Scientific, Waltham, MA) was then added into the base mold until it slightly touched the dorsal (down) side of the brain mold. The freezing platform, with the base and brain mold still attached, was placed in a $-80^{\circ C}$ freezer until the Neg50 was solid.

When the Neg50 was fully frozen, the brain mold was briefly thawed by a heat gun to remove it from the base mold. The surface temperature of the brain mold cavity was kept at $-2^{\circ C}$ to hold the brain shape while leaving the Neg50 solid. Additional Neg50 was then added to the base mold, filled to a volume to sufficiently immerse the brain, and left to thermally stabilize for 15s. The brain was removed from the 30% sucrose in 0.1M PB solution and dried for 30–45s before being carefully placed within the Neg50 filled base mold with the ventral side of the brain facing up at a 0° horizontal plane. The base mold was then placed in dry-ice

chilled 2-methylbutane until all the Neg50 was uniformly frozen. Finally, the base mold was thawed by a heat gun to remove the brain block from the base mold apparatus, placed in a properly labeled freezer bag, and stored in a $-80^{°C}$ freezer.

# Appendix 4

DOI: https://doi.org/10.7554/eLife.40042.016

## Cryo-sectioning

The cryostat's stage was modified to accommodate the larger dimensions of a cryo-embedded brain block and aided in stabilizing the cryostat's chuck and blade (*Appendix 4—figures 1*). The UV-LED device was arranged in 4 rows of 11 LEDs in a parallel resistor network to provide uniform UV intensity across the surface of the slides. Each array was connected to a single 6V DC power source and regulated by an on-off timer controller using a Raspberry Pi 3 (*Raspberry Pi foundation, 2016*). *Figure 2c* shows the setup of the UV station within the cryostat.

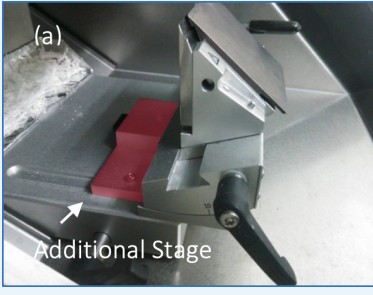 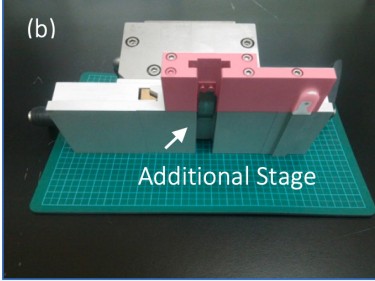 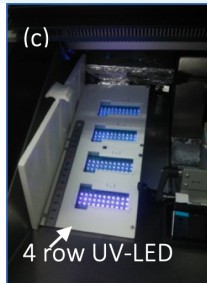

**Appendix 4—figure 1.** Modified cryostat chamber to accommodate for larger brain block. (**a,b**) Additional stage shown in pink was attached to the original cryostat stage to increase the room space and to aid in stabilization of the cryostats chuck and blade. (**c**) a four row UV-LED device to provide UV intensity across the surface of the slides by an on-off timer controller using a Raspberry Pi.

DOI: https://doi.org/10.7554/eLife.40042.021

## Appendix 5

DOI: https://doi.org/10.7554/eLife.40042.016

### Histology (Staining)

The slides for **Nissl** staining were processed through an automated Nissl staining protocol beginning with a thionin solution: 1.88 g thionin chloride (TCI, T0214) in 750 mL De-ionized $H_2O$ (Di$H_2O$), 9 mL of glacial acetic acid (WAKO, 012–00245), and 1.08 g sodium hydroxide pellets (Sigma-Aldrich, 221465–500G). The slides then underwent three washes of Di$H_2O$ followed by dehydration in increasing concentrations of ethanol 50%, 70%, 95%–100% and finally xylene, (**Nissl, 1894**; **Pilati et al., 2008**) followed by automatic cover-slipping.

The **myelin** staining technique used a modified ammoniacal silver impregnation technique originally developed by Gallyas (**Gallyas, 1979**). Instead of the standard protocol of implementing the technique on free floating sections, the protocol was applied to the slide mounted sections. After the physical development of the myelin stain, the tissue was manually inspected for staining and morphological quality. The slides were then put on a drying rack for 24 hr and were dehydrated with ethanol followed by automatic cover-slipping.

The **CTB** designated slides were manually loaded into Immunohistochemistry (IHC) basins (Light Labs, LM920-1). The basins were filled to ½ of their volume with tap water to maintain humidity levels. In our CTB-DAB (3,3'diaminobenzidine) protocol, the solutions were pipetted with ~800 µL onto each slide. The first step, blocking, consisted of 15 mL methanol (Nacalai Tesque, 21915–35), 480 mL 1xPBS, and 1.25 mL hydrogen peroxide $H_2O_2$ (Wako, 081–04215) was shaken and sprayed onto the slides. The protein block was made of 1% v/v triton X-100 (Sigma Aldrich, X100-500G), 3.5% v/v normal rabbit serum (Vector Labs, S-5000) in 1xPBS for 30 min at room temperature (RT), followed by 1xPBS rinse 3 times and pressure assisted drying. The primary antibody step consisted of 2% v/v goat anti-CTB (List Laboratories, #703) (1:2500 concentration), 0.3% v/v triton X-100, 3% v/v normal rabbit serum in 1xPBS which was left overnight at RT with the IHC basins covered to preserve liquid levels and ambient humidity within the basin. Once the slides went through a 1xPBS rinse 3 times and pressure assisted drying, the secondary antibody made up of 0.4% v/v biotinylated rabbit-anti-goat IgG (H + L) (Vector Labs, BA-5000) (1:250 concentration), 1% v/v normal rabbit serum, 0.3% v/v triton X-100 in 1xPBS was added and left for two hours at RT. After another 1xPBS rinse 3 times and dry cycle the Avidin-Biotin Complex Elite Kit (ABC, Vector Labs, VEC-PK-6100) was placed on the slides and left to incubate for three hours at RT. The ABC kit was used with equal volumes of avidin and biotin, 1% v/v avidin and biotin were made 30 min before use.

Our DAB-Nickel Cobalt (DAB-NiCo) staining protocol used 1% w/v DAB (Apollo Scientific Limited, BID2042) 1% w/v ammonium nickel (II) sulfate hexahydrate (Santa Cruz Biotechnology, sc-239235), 1% w/v Cobalt (II) Chloride hexahydrate (Sigma Aldrich, 255599–500G), and 0.00003% v/v $H_2O_2$. The DAB, Ni, Co and $H_2O_2$ were prepared with a Di$H_2O$ in 50 mL conical tubes and filled to 50 mL. 150 µl of hydrochloric acid (Nacalai Tesque, 18320–15) was added to the 50 mL conical DAB tube to ensure homogeneity. 800 mL of 1xPBS was prepared then added to a 2L Erlenmeyer flask placed on a stir plate, the 1% DAB-NiCo solutions were added to the Erlenmeyer flask, and was homogenized with a stir rod. 350 µL of 10M NaOH (AppliChem, A3910,1000) was added to bring the final pH of the DAB-NiCo to 7.1–7.4 pH.

A glass basin large enough to contain 1L of liquid was used inside a fume hood and all slides within the IHC basins were manually loaded into slide racks and placed within the basin. The $H_2O_2$ was added to the flask just before staining to catalyze the DAB-NiCo reaction. The final working solution was poured from the flask into the basin where the slides had been placed. The incubation time (~10 min) was monitored manually until the injection site could be visualized as the affected cells turned black. Manual monitoring was used to make sure that the signal-to-background noise ratio was kept from being deleterious to the final stain quality. The slides were then transferred through three full emersion washes of 1xPBS.

The slides were left on a drying rack overnight at RT and were put through a Giemsa counterstain after a 24 hr period. The Giemsa counterstain consisted of a 3:7 ratio of 30% Giemsa (Nacalai Tesque, 37114–35) and 70% DiH$_2$O, a 1xPBS wash, 1% w/v ammonium molybdate (Sigma-Aldrich, A1343-100G) wash, a second 1xPBS wash followed with ethanol dehydration. The slides were then cover-slipped and put into drying racks for 24 hrs.

## Appendix 6

DOI: https://doi.org/10.7554/eLife.40042.016

# Computational Infrastructure

All data machines within the laboratory were connected to data center using a 10G network for further analysis by a 16 node high-performance computing (HPC) cluster. Storage nodes were configured as Raid6 devices and provided 78TB of useable disk space each for a total of 156TB. The theoretical maximum transfer rate of the 10G network is 900 MB/s; however, the rate limiting process was due to the hard disk writing speed of each machine. The current average transfer speed is about 250–520 MB/s.

The data processing (cropping and converting) in the data-acquisition server began when the Nanozoomer slide scanning was completed. The processed data were transferred to a central repository for quality control. This configuration could improve the overall process rate from 50% of the theoretical maximum up to 80% in performance. As shown in *Appendix 6—figure 1*, an entire network and computation pipeline setup was adopted in the RIKEN marmoset Neural Circuit Architecture Laboratory.

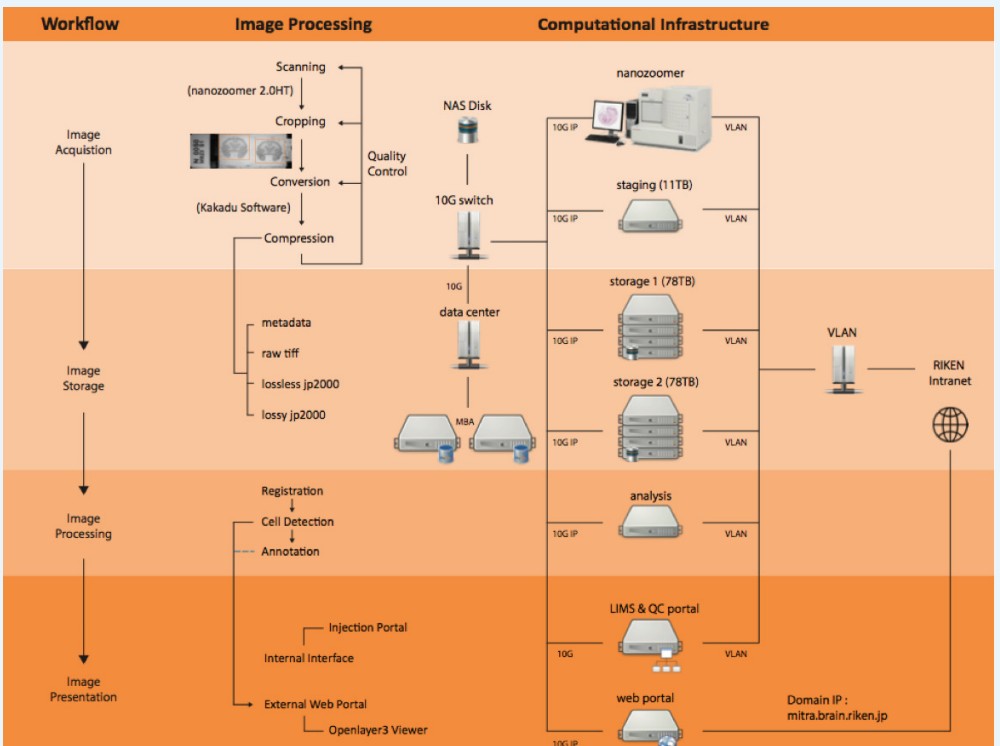

**Appendix 6—figure 1.** Computational pipeline with the network structure to perform a high-throughput data flow and process. There were four steps of workflow involved in this pipeline including image acquisition, storage, processing and presentation. With these steps, generating a whole marmoset brain dataset with high production rate and superior system performance for large data communication was possible. Each server node was connected to one 10G network for data communication and one external network for remote access.

DOI: https://doi.org/10.7554/eLife.40042.024

## Appendix 7

DOI: https://doi.org/10.7554/eLife.40042.016

## Computational Processing

The computational pipeline builds upon the pipeline originally developed by the CSHL Mitra Lab for the mouse and was modified to meet the marmoset tissue size and structure (*Appendix 7—figure 1*). For each tissue section, the system produced (1) a meta-data file with all the relevant information (cropping and conversion processing); (2) a cropped ROI as a TIF format for image inspection; (3) a down sampled JPEG2000 image for rapid access for the data on the web portal; (4) an uncompressed raw data file. Our automatic detection algorithm for the cropping box placement in images performs at a 100% success rate in both brightfield and fluorescence sections. The image format used custom scripts based on the Kakadu toolkit (*Kakadu, 2016*). For any given complete marmoset brain, there were a total of ~1700 sections mounted on ~ 900 slides.

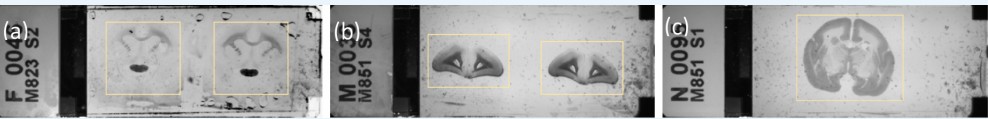

**Appendix 7—figure 1.** Example of a Nanozoomer macro image determining the cropping ROIs. (**a**) fluorescence slide (**b and c**) brightfield slides shown with yellow cropping box.
DOI: https://doi.org/10.7554/eLife.40042.026

This project developed and utilised an online quality control (QC) service. The QC service was employed to assess the quality of each image and determine if the re-imaging of a slide or the re-injection of an entire brain was needed. Correcting or improving the pipeline process was an evolutionary and organic process and flagging unwanted sections or materials to reduce unnecessary post-processing was a key step. The researcher had the option to view all the sections of the series (with the comparison of a micro image) and to edit the fields for QC such as tissue damage, missing sections and poor cover-slipping alignment. Once flagged, the QC service would automatically remove the sections from the dataset allowing for proper processing of the image analysis such as 2D alignment and 3D reconstruction.

The images of brain slices from histological processing were fed directly into the computational process. The post processing data involved several steps from image cropping and image conversion to 2D alignment and 3D reconstruction. This was a major departure for image analysis. Acquired image datasets were written into a propriety image format, JPEG2000. In the case of a JPEG2000 image, the decompression was $\approx$ 75–90 MB for fluorescence images and brightfield images (Nissl, myelin and CTB). After the proper data acquisition, automatic image processing/analysis was performed. The sections across all brains were registered into a common space. This registration was based on Nissl-stained sections for structural information and ex-vivo MRI as landmarks such that all sections were able to align to each other and produce a shape similar to that of the same subject reference (ex-vivo MRI) while maintaining coherence and continuity from section to section. A variant of the large deformation diffeomorphic metric mapping (LDDMM) algorithm was employed to compute nonlinear mapping between Brain/MINDS Nissl atlas and the reconstructed target Nissl, followed by recently developed registration methods (*Lee et al., 2018*).

# Appendix 8

DOI: https://doi.org/10.7554/eLife.40042.016

## Processing Rate

Based on the individual marmoset brain anterior-posterior length measured by ex-vivo MRI, the number of sections (20 μm/section) was determined and the processing time at each step was recorded. At each step of histological processing, a small portion of brain sections were excluded from the subsequent processing based on manual quality control inspection. The final processing success rate for each series was measured by the percentage of obtained sections, shown in *Appendix 8—figure 1*.

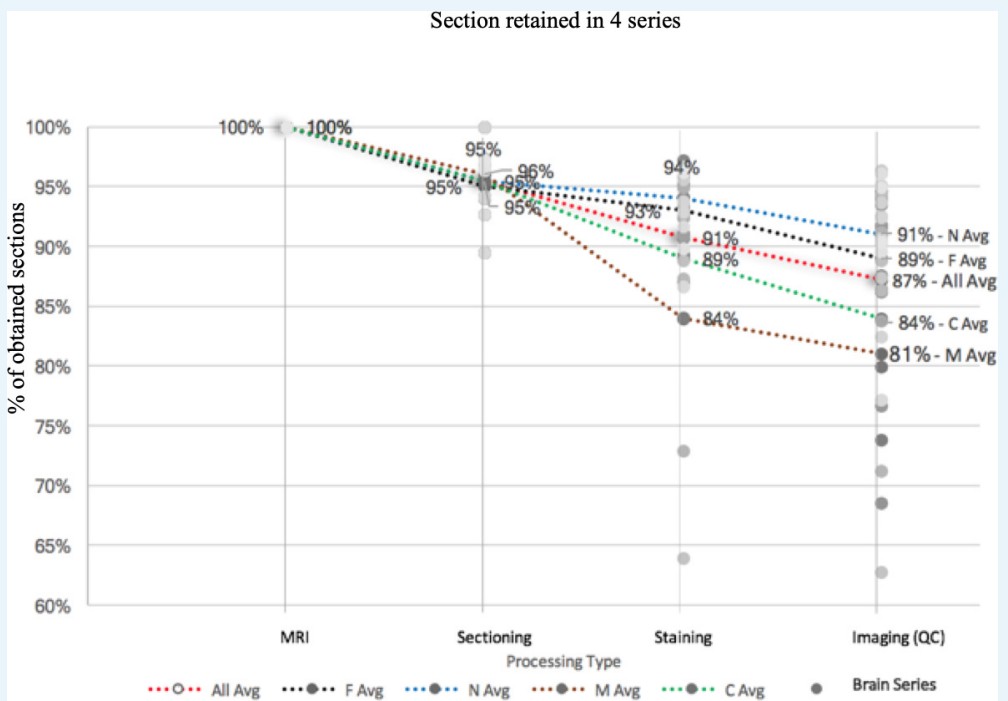

**Appendix 8—figure 1.** A pipeline processing rate with four series staining (fluorescence, Nissl, myelin, CTB) based on the latest 10 datasets. Each series starts with 100% full rate (based on the calculation from ex-vivo MRI and the number of sections needed as well as calculated by measuring at 20 μm each) and reduces by a percentage based on unavoidable reasons such as poor staining or section peeling. The figure shows that there is high processing rate starting with Nissl (91%), fluorescence (89%), CTB (84%), to myelin (81%). The average processing rate is 87% in total.

DOI: https://doi.org/10.7554/eLife.40042.028

## Appendix 9

DOI: https://doi.org/10.7554/eLife.40042.016

## Grid-based Approach and Plan for Whole-brain Mesoscale Circuit Mapping

### A. Adaptive Grid-based injection plan: distribution of injection targets over the grey matter.

In order to obtain a data set addressing the question of whole-brain mesoscale connectivity, we adopted the approach of injecting on a systematic grid-based plan throughout the whole brain's grey matter. The grid-based approach was originally established for the Mouse Brain Architecture project, with an initial regular grid with 1 mm injection spacing, adapted to avoid region boundaries (*Grange and Mitra, 2011*). The marmoset brain is larger than the mouse brain and a similarly dense set of injections would be prohibitive to carry out in terms of numbers of animals required. We therefore expanded the grid spacing to 2 mm. Note that our plan is not strictly a regular grid as such a grid would also generally overlap with region boundaries. We started with an initial regular grid, then shifted/adapted grid points to avoid boundaries.

The process is not entirely automated, as the atlas compartments are heterogeneous, and we adapted injection placement by actual visualization of the compartment volumes. In addition, there is also individual variability between brains, and injections themselves have size variations (e.g. due to differential fluid transport properties in different brain regions). We addressed the question of individual variation in part using in-vivo MRI guidance of injections, particularly for sub-cortical nuclei. We also tried to ensure that the injection surgeries were performed by expert neuroanatomists with knowledge of stereotactic injections in the specifically targeted regions of the marmoset brain. In this way we attempted to perform better than a regular geometrical grid and accounted partially for animal-to-animal variations.

For grid planning we adapted the Paxinos/Hashikawa atlas as a starting template (*Hashikawa et al., 2015*; *Hikishima et al., 2011*; *Woodward et al., 2018*), and initially placed a 3D 2 mm grid covering all the grey matter areas, which yielded 271 injection centers in one hemisphere. This analysis missed some of the smaller structures which have volumes less than 8 mm$^3$. However, some structures in the reference atlas are very small and not practical to inject. To determine a size threshold for future planning purposes we examined the actual sizes of the injections placed. Accounting for size variations in the tracer injections, we have found that a diameter of injection spread diameter as small as 0.8 mm and as large as 2.5 mm (assuming spherical spread; see analysis below) could be achieved in practice. We therefore could plan at least one injection center for each brain region with volume ranging from 0.27 mm$^3$ to around the grid size of 8 mm$^3$. *Appendix 9—figure 1* presents the total number of injections, and the number of animals need to be involved in the experiments, with regard to different sizes of injection spread.

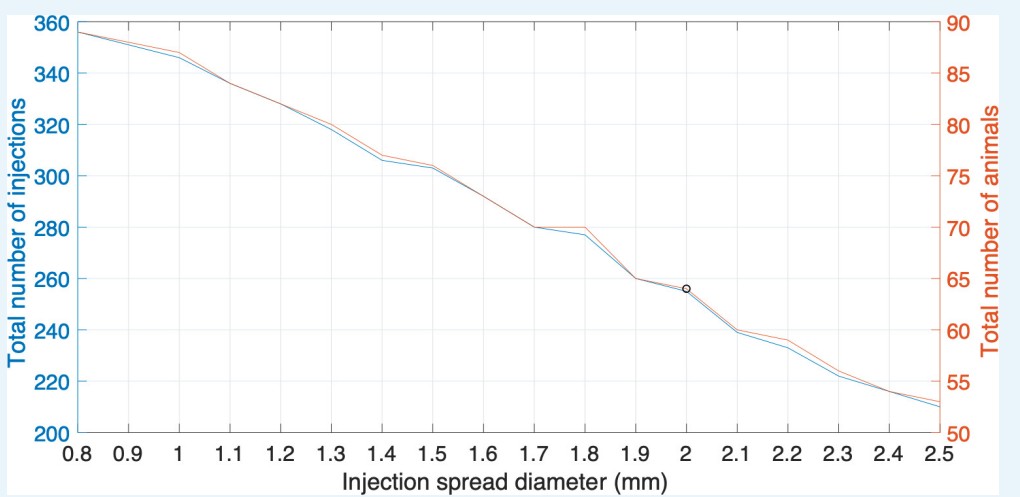

**Appendix 9—figure 1.** Plots representing the total number of injections, the number of animals needed and injection spread diameter (mm). Each plot represents the different sizes of injection spread in diameter (mm); the right side y-axis represents the total number of animals required to be involved in the experiments; the left side y-axis represents the total number of injections. The black circle is the cutoff where a 2 mm diameter injection spread requires 255 tracer injections throughout a total of 64 marmoset brains. The cutoff represents a reasonable balance between minimal animal use versus maximum number of tracers that can be used in this experiment.

DOI: https://doi.org/10.7554/eLife.40042.030

Assuming a spherical injection spread with 2 mm diameter, our overall 'adaptive' grid plan for injections contains a total of 255 injection centers in 241 target structures in one hemisphere's grey matter (*Appendix 9—figure 1*). Since each site in the plan is separately targeted with retrograde and anterograde tracers, this implies a total placement of 510 injections. In our plan we try to maximize the number of tracers per animal, to minimize animal number, by placing 2 retrograde and two anterograde injections. Thus, the placement of 510 injections requires 64 marmosets. It is necessary to prioritize the larger areas (and the areas that will be less failure prone when injected), and also ideally to combine data across groups.

Given the considerations as described above, in the cerebral cortex, 398 injection centers cover 118 target structures, comprising 74% of the total grey matter volume (*Appendix 9—figure 1*). The largest regions such as V1 and V2 contain 33 and 15 injection centers, respectively, while small regions such as anterior intraparietal area (AIP) and temporal area 1 (TE1) have only one injection center each.

**Protocol note:** in-vivo MRI was performed on every animal before tracer injection to obtain *a priori* information of the individual brain's anatomy. Using the approach of Large Deformation Diffeomorphic Metric Mapping (LDDMM) (*Ceritoglu et al., 2010*), the marmoset brain atlas is matched to the individual's brain MRI images so as to provide guidance on the injections (see Section 2.2). In addition, for specific subcortical injections we adopted an in-vivo MRI based stereotaxic surgery procedure to ensure accurate placement of the injections in subcortical nuclei (*Mundinano et al., 2016*).

It is to be noted, that despite best efforts, it is impossible to guarantee that every injection is placed within the center of a regional boundary as designated by a reference atlas. However, we do not regard this as a fundamental obstacle to obtaining a draft connectivity map of the marmoset brain. Our approach is conservative in that it uses far fewer animals than would be required if there was an insistence of precise placement of the injections, as this more conservative approach is inherently more lossy as well as costly in terms of animals. Secondly, the regional boundaries are themselves open to discussion and debate. Ultimately these debates need to be settled by the availability of unbiased brain-wide data sets and using an existing reference atlas to precisely place injections may occasionally perpetuate

previous errors. Thirdly, our data sets are comprehensive 3D brain-wide volumes with multiple histological series, permitting both computational analysis, and expert neuroanatomists to draw their own judgments.

We feel therefore that the grid-based approach, while imperfect, is an important stepping stone towards understanding marmoset brain circuits in particular and primate brain circuitry in general. It is without doubt the case that the current data set constitutes a major advance in this area. For details on target structures and number of injections, please refer to *Supplementary file 1*.

## B. Grid coverage in the current data set, and considerations of tracer spread across regions

As of the publication of this paper, we have placed 178 injections in the cerebral cortex, including both anterograde and retrograde tracer injections (see Section 2.1), covering 47% of the planned injection centers. 12 injections over three injection centers have been placed in the thalamus. 47 brains including 99 injections in the cerebral cortex have been processed through the experimental and computational pipeline. Manual annotation was performed to assess the fidelity of injection to the plan. 73% of the injections were restrained within the relevant anatomical boundary, while 27% injections had tracer leakage into adjacent regions. About 21% of the injections restrained within one region were within the large cortical regions including V1, V2 and V3. Among the injections with tracer spread into more than one region, about 18% (5 injections) were due to actual injection centers placed too close to the anatomical boundary. Within a sample dataset of 15 injections in V1, V2 and V6 from eight animals, we assessed the extent of tracer injection. For simplicity, we assumed a spherical spread of each tracer injection. The diameter of the injection extent ranged from 0.82 mm to 2.46 mm. When discounted by the variation in tracer volume, the diameter of injection extent based on 0.3 µl tracer volume ranged from 0.59 mm to 2.46 mm, with medians of 2.15 mm for AAV-GFP (n = 5), 1.60 mm for AAV-tdTom (n = 4), and 1.93 mm for FB (n = 6). Among these 15 injections, 6 of those had tracer spread beyond the anatomical boundary. On average, for each tracer spread, about 68% of the volume was restrained within the same region as the injection center, while about 32% of the volume leaked outside to adjacent regions.

## C. Future plan

The adaptive grid plan based on the Paxinos/Hashikawa atlas, with a cutoff 2 mm diameter injection spread, requires 510 tracer injections (anterograde + retrograde) throughout the marmoset brain, including 398 in the cerebral cortex. A total of 190 injections have been placed to date in 49 animals, 178 in the cortex and 12 in the thalamus. To cover the rest of the brain with the cutoff as indicated above, 220 more injections would need to be placed in the cortex, and 100 injections would need to be placed in subcortical regions and cerebellum. This would require a total of 80 more marmosets. The high-throughput pipeline presented in the paper has a capacity of 2 marmoset brains/month but this number may be scaled up by replicating equipment (the scanning being a rate limiting step). If the current rate were maintained the plan could be completed by 2022. However, by prioritizing the larger areas, the project could be completed more quickly and with fewer animals. The *Appendix 9— figure 1* could be utilized to adapt injection/animal numbers based on an injection cutoff.

## Appendix 10

DOI: https://doi.org/10.7554/eLife.40042.016

# Individual Variation and Impact on Injection-based Projection Mapping

## A. Individual variation of marmoset brains: compartment sizes

Any brain connectivity mapping approach should address the question of individual variation. Notably, previous work on large scale mesoscale connectivity mapping has been carried out in the C57BL/6 male mouse strain controlled for age and weight. Such an approach is impractical for a primate given the numbers involved. While it is not possible to tightly control age and weight (and perform many repeats), we can still assess the extent of individual variations. Note that as our injections are tailored based on an in-vivo MRI in the same animal, overall size variations are controlled for to some extent.

We would also note that the stereotactic reference histological atlases used in previous studies are for an individual animal, and no real attempt has been made in the literature, to *explicitly* study the effects of individual variation on the reference atlas. In some instances, multiple brains are averaged to produce a smoother reference brain, but this does not explicitly address the issue of individual variation – it is as if in a multivariate distribution, the *mean vector* was given, but *not the covariance matrix*! In this case, not even basic statistical analysis is possible. This is a prevalent problem in the literature and it would be too much to resolve in the current study. Nevertheless, we possess a uniquely large 3D histological data set together with in-vivo and ex-vivo MRI that will permit an unprecedented study of brain compartment size variations. We briefly commence that study here and will pursue in more detail in a future publication.

To account for the volumetric variation across different marmoset brains, we calculated the volumes of the whole brain based on in vivo MRI results. Note that our marmosets are mostly female; we did not have a large enough male sample to systematically assess gender differences. Within a sample of 26 cases, the whole brain volume had a median of 8222.5 mm$^3$ with a median absolute deviation (MAD) of 319.4 mm$^3$. In comparison to the Paxinos/Hasikawa (Brain/MINDS) template (*Hashikawa et al., 2015*; *Hikishima et al., 2011*; *Woodward et al., 2018*), our animals were older and mostly heavier than the template brain animal. Yet the brain sizes were similar to the template brain. We did not find a significant relationship between whole brain volume and age or body weight (*Appendix 10—figure 1*) within our data set.

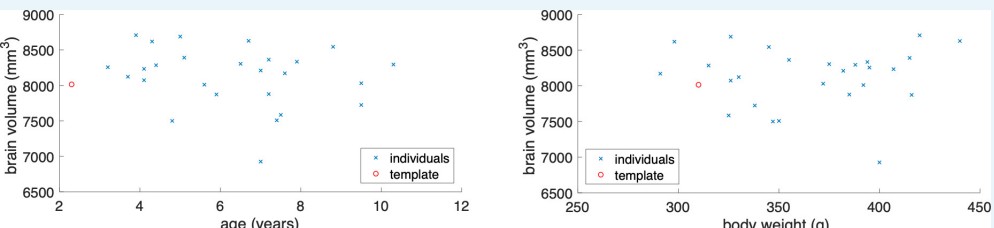

**Appendix 10—figure 1.** Relationship between whole brain volume and age or body weight. The left plot shows individual marmoset variation between whole brain volume and age in comparison to the Brain/MINDS template. The right plot shows individual marmoset variation in comparison to body weight and to the Brain/MINDs template. The red circle represents the Brain/MINDs template brain and the blue crosses represents individual animals in this experiment. These plots show no significant relationship between the template brain and individual experimented brains.

DOI: https://doi.org/10.7554/eLife.40042.032

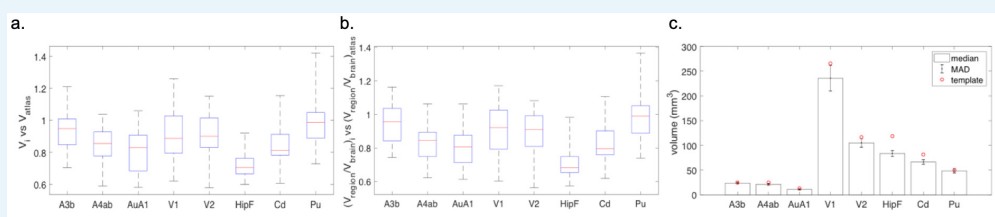

**Appendix 10—figure 2.** Comparison of individual variability of representative brain regions against the template brain. (**a**) Box plots of ratio of each brain region's volume in individual animals against its volume in the template brain, where the red line shows the median, the lower and upper bound of the box shows the 25th and 75th percentile data, respectively, and the whiskers extend to most extreme data points. A ratio of 1 means the same volume between the brain region in the animal(s) involved in the current project and the template brain. A ratio lower/higher than one means smaller/larger brain region in the animal in the current project compared with the template brain. (**b**) Box plots of each brain region's proportion in the entire brain in individual animals against the proportion in the template brain. Similar to (**a**), the red line shows the median, the upper and lower bound of box shows the 75th and 25th percentile data, and the whiskers show the most extreme data. A ratio of 1 means the same proportion of the brain in the individual compared with the template brain. (**c**) Bar plot of the absolute volume of individual brain regions across different animals. Height of the bar shows the median and the error bars show the MAD.

DOI: https://doi.org/10.7554/eLife.40042.033

To address the variation of specific brain compartments across animals, we estimated the volumes of individual anatomical regions based on the MRI-guided atlas mapping (see Section 2.2). By mapping the template brain regions to individual brains, we compared the results from 23 samples with the reference atlas and presented example regions including A3b (primary somatosensory cortex), A4ab (primary motor cortex), AuA1 (primary auditory cortex), V1 (primary visual cortex), V2 (secondary visual cortex), HipF (hippocampal formation), Cd (caudate nucleus) and Pu (putamen). The absolute volumes of these regions (median ± MAD) are also presented. Data is provided in *Appendix 10—table 1*.

**Appendix 10—table 1.** Median and MAD of each metrics evaluating the brain region volume's variability across animals. The table shows some of the large components in the marmoset brain.

| | | Whole brain | 'A3b' | 'A4ab' | 'Aua1' | 'V1' | 'V2' | 'Hipf' | 'Cd' | 'Pu' |
|---|---|---|---|---|---|---|---|---|---|---|
| $V_i$ | Median | 8234 | 23.76 | 21.40 | 10.91 | 235.76 | 105.04 | 83.64 | 66.26 | 48.55 |
| | MAD | 351 | 2.38 | 2.46 | 1.41 | 36.70 | 12.71 | 7.42 | 7.75 | 5.93 |
| $V_i/V_{atlas}$ | median | 1.03 | 0.95 | 0.86 | 0.83 | 0.89 | 0.90 | 0.71 | 0.81 | 0.99 |
| | MAD | 0.04 | 0.09 | 0.10 | 0.11 | 0.14 | 0.11 | 0.06 | 0.10 | 0.12 |
| $(V/V_{brain})_i/(V/V_{brain})_{atlas}$ | median | 1 | 0.96 | 0.85 | 0.81 | 0.92 | 0.91 | 0.68 | 0.80 | 0.99 |
| | MAD | 0 | 0.10 | 0.09 | 0.10 | 0.12 | 0.10 | 0.07 | 0.09 | 0.11 |

DOI: https://doi.org/10.7554/eLife.40042.034

In the context of generating population-based atlas, previous studies in humans (*Yeh et al., 2018*) (and non-human primates (*Black et al., 2001a*; *Black et al., 2001b*; *Feng et al., 2017*; *Hikishima et al., 2011*; *Quallo et al., 2010*) mostly focused on mapping individual brains to a common template. Individual variations were addressed in terms of variation in stereotaxic coordinates of major landmarks such as sulci and caudate (*Black et al., 2001a*; *Black et al., 2001b*; *Hikishima et al., 2011*). Few studies explicitly reported the variations in brain sizes involved in their studies (*Hikishima et al., 2011*). No study thus far has completed a region-based variation comparison as in this paper, and no

study has looked at the multivariate covariance between structures. We will address these questions in a future publication.

From the considerations above, it is clear that there is both significant variation in the absolute compartment volumes, and in the relative volumes normalized by the whole brain, of individual marmosets compared to the reference atlas. Thus, individual variation is present and cannot be ignored. Nevertheless, we feel that the traditional method of repeating the same injection many times in the same region, is not practical for the brain-wide connectivity mapping using the present approach. We utilize three primary tools to address questions of individual variation. First, we perform in-vivo MRI to ensure injection placement within the compartments of choice. Second, we use 3D histological series and diffeomorphic atlas mapping, in order to quantify the precise placement of each injection and corresponding projections, according to a mapped reference atlas. Third, as discussed below, it is possible to combine injections with data sets gathered by other investigators, to virtually increase the sample size. This is particularly feasible if the other data sets are also available in 3D atlas mapped form. This is discussed further in the next subsection. This approach of combining our project injections with data from other investigators has already led to collaborative publications (*Huo et al., 2018*; *Majka et al., 2018*).

## B. Combining with other tracing studies

To gain an understanding of the possibilities of combining project data with data from other investigators, we compared five injection locations in four anatomical regions where the injections from the current pipeline and the ones previously gathered in the Rosa lab were in close proximity as evaluated by the stereotactic coordinates of the injection centers (distance ranging from 0.8 to 2 mm) (see http://marmoset.brainarchitecture.org for all brains referred to here). All injection extents were restrained within the same brain regions as the injection centers. *Appendix 10—figure 3* shows transverse projection of the injection locations. It is clear, that there are examples of injections that can be combined/compared across the projects. We will pursue such a combination/comparison study in a future publication.

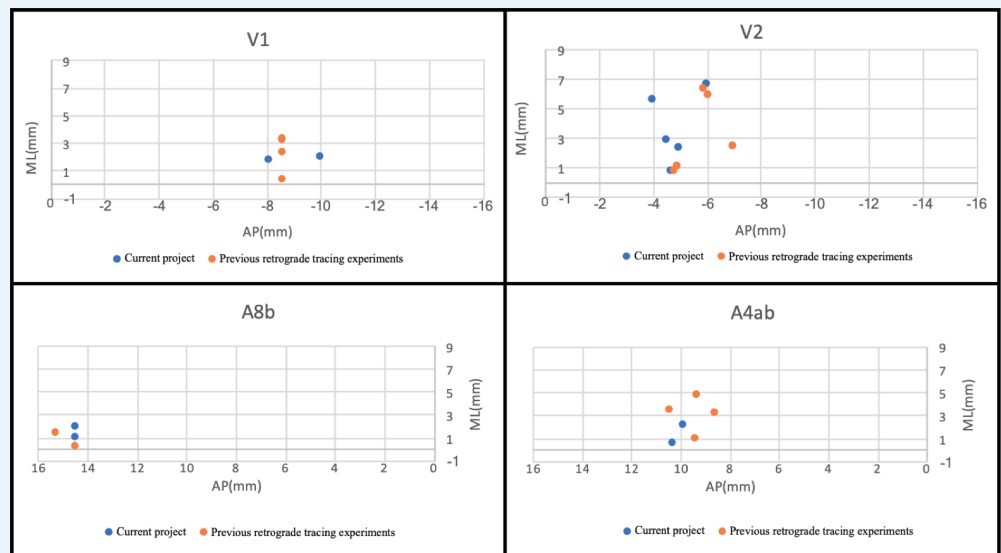

**Appendix 10—figure 3.** Transverse projection of the injection locations between individual brains in the current project and previous retrograde tracing experiments. The similar plots (injections) presented here in V1, V2, A8b, and A4ab suggests that the current grid method is feasible and can be further analyzed across other collaborative projects.
DOI: https://doi.org/10.7554/eLife.40042.035

