## [Decision Letter]

Thank you for submitting your article "A High-throughput Neurohistological Pipeline for Brain-Wide Mesoscale Connectivity Mapping of the Common Marmoset" for consideration by *eLife*. Your article has been reviewed by three peer reviewers, including Moritz Helmstaedter as the Reviewing Editor and Reviewer #3, and the evaluation has been overseen by Eve Marder as the Senior Editor. The following individual involved in review of your submission has agreed to reveal their identity: Trichur R Vidyasagar (Reviewer #2).

The reviewers have discussed the reviews with one another and the Reviewing Editor has drafted this decision to help you prepare a revised submission.

This is a relevant methodological contribution to extend methods for mesoscopic connectomic analysis developed in mouse to a non-human primate, the marmoset. The reviewers and editors consider this a potentially important set of tools, and the prospect of high-resolution projection data from the marmoset brain inspiring.

However, they agreed to request the following addition/edits to the manuscript:

– a more thorough treatment of the issue of variability, injection granularity (smaller is better for connectivity precision, but becomes intractable when too many distinct injection sites are chosen), and realism of time scales of data acquisition.

– How is this issue treated in the context of inter-animal variability? Please provide quantitative estimates.

– clear proof that achieving a marmoset connectome with the reported technique is feasible

These points may require more data, or simply a thorough quantitative discussion based on the data provided.

The remaining points raised by the reviewers can be treated as recommendations for the revised manuscript.

Reviewer #1:

This is a detailed technical account of a process that allows high-resolution tracer injection and mapping in the brain. The mechanics of the process described are standardized, with full details of components used, presumably to ensure high reproducibility, and it is referred to as a pipeline.

There is a wealth of details, which I presume will help replication of the pipeline and I expect that this is the main interest of the paper. Overall the paper describes a near industrial approach to obtaining high fidelity mapping in a highly systematic fashion. The validity of the paper is not addressed, contrary to what is suggested in the Abstract where it states that 'we are able to overcome individual variation exhibited by marmosets to obtain routine and high quality maps to a common atlas framework'. This misleading sentence could suggest that brain variability in terms of areal layouts or connectivity have been addressed in the study. This is not the case.

The paper is not a study in the usual sense, nevertheless it is subdivided into Introduction, Materials and methods, Results and Discussion. None of these elements have a common bearing. For instance the Introduction has a long section on the phylogenetic tree, which might lead one to expect that the paper deals with marmosets. It doesn't. Marmoset brains are processed in the pipeline but we learn nothing about them that would warrant understanding their position in evolution. Likewise, the Introduction touches on curated databases and develop at length the Felleman and Van Essen, 1991 study leading to the work that established the CoCoMac data base. This might lead one to expect that the paper would lead to data and that the merits and otherwise of curated data would be relevant. That neither is the case.

My point is that the Introduction could be scrapped with no loss to the paper; Idem for the Discussion. If this paper were to be a study, various parameters of the pipeline would be modulated so as to see how this impacts the results. For example, if one is to embark on a grid approach to determining the connectome (as is the case here) what is the optimal frequency to be used? Clearly this is related to brain size. If one is to use a template approach, as in the present study, what impact does that have on the design of the pipeline and what are the controls that would allow one to isolate the template effect? What are the short-comings of a grid approach? What are the short-comings of a template approach? These are burning issues facing connectomics today that are not touched upon by this methodological paper.

Reviewer #2:

Achieving a comprehensive connectivity map of the primate brain at the 'mesoscale' is a challenge and the authors are to be commended for taking it on. The manuscript is essentially the description of a method that the authors have designed and tested for its feasibility. It requires high throughput, both experimentally and computationally. The manuscript has indeed described a feasible way of achieving the aim of a whole-brain connectivity map of the marmoset. This can potentially pave the way for applying the same techniques to larger brains such as the macaque's.

I see three major challenges in the ultimate objective for producing a useful connectivity map for the marmoset:

1) How well can the size of injections be controlled? Spread to neighbouring areas will be a problem. However, with greater number of animals and appropriate computational tools, I agree that one can improve the results.

2) Smaller the site of injection, better will be the final resolution of the connectivity map. However, that will increase the number of injections that need to be made.

3) There will be animal-to-animal variability and with the marmoset, I believe one does not have the same degree of luxury of having precisely the same genetic strain as with mice. This might require the experiments to be repeated with a large number of animals. I am not sure that "access to high-quality in-vivo and ex-vivo MRI" as stated in the manuscript will be adequate.

I would have liked to have seen some discussion about realistic numbers of animals and injections and the time needed for the project to be completed. I guess this would require a large consortium of laboratories. Some plausible projections would add to the feasibility of this ambitious programme.

Reviewer #3:

This is an important, relevant, and well carried out methodological contribution. While systematic projectional analyses of mouse brains have become available since 2014, the application and extension of such approaches to larger brains, and most notably a non-human primate brain, is of substantial relevance for obtaining comparative data on large-scale brain connectivity. I particularly commend the authors on Figures 1 (evolutionary context) and Figure 7 (detailed explanation and illustration of the connectivity matrix generation).

Importantly, the data viewer should be operational as advertised. Currently, http://riken.marmoset.brainarchitecture.org/ doesn't load, and shows black data only. Please assure that the data browser is operational at review stage, since this is one important aspect of the presented methods.

---

## [Author Response]

This is a relevant methodological contribution to extend methods for mesoscopic connectomic analysis developed in mouse to a non-human primate, the marmoset. The reviewers and editors consider this a potentially important set of tools, and the prospect of high-resolution projection data from the marmoset brain inspiring.However, they agreed to request the following addition/edits to the manuscript:– a more thorough treatment of the issue of variability, injection granularity (smaller is better for connectivity precision, but becomes intractable when too many distinct injection sites are chosen), and realism of time scales of data acquisition.

Our goal is to generate a brain-wide mesoscale connectivity map in the marmoset a reasonable time period, which we believe is of value given the lack of ground-truth (ie tracer-based) whole-brain projection data in any primate species. As such, it is impractical to repeat every injection many times as is the case in hypothesis-driven studies where the focus may be on a specific brain region or a pair of regions. This would require too many animals. Nevertheless, we do nominally have N=2 as the anterograde and retrograde injections when combined can provide confirmation of a given projection. In addition, we place the stereotactic injections using in-vivo MRI guidance and a reference atlas, and are able to better target brain structures delineated in the atlas atlas. We have carried out an analysis of the compartment size variability in our data set, and include the results in a new appendix section (Appendix 10).

We also treat the issue of injection size variations in a new appendix section (9), where we provide the analysis of the grid-plan called for by the reviewers, including estimates for the numbers of animals required to complete the study in a reasonable period of time. In addition to these appendix sections, we have also included two new subsections in the manuscript (i) on individual variability (ii) the complete grid-based plan, together with completion scenarios and resource requirements.

– How is this issue treated in the context of inter-animal variability? Please provide quantitative estimates.

An appendix section (10) has been provided on the issue of inter-animal variability, also see discussion above.

– clear proof that achieving a marmoset connectome with the reported technique is feasible

An appendix section (9) is provided with the complete grid plan as well as estimates of the resources required to complete the project. Together with our current progress, these estimates demonstrate the feasibility of completing the brain-wide coverage in a few year’s time, which we regard as inevitable given the scale of the challenge.

These points may require more data, or simply a thorough quantitative discussion based on the data provided.

We continue to gather more data, however our response to the reviewers is based on a more detailed analysis of our existing data as well as some literature review.

The remaining points raised by the reviewers can be treated as recommendations for the revised manuscript.Reviewer #1:This is a detailed technical account of a process that allows high-resolution tracer injection and mapping in the brain. The mechanics of the process described are standardized, with full details of components used, presumably to ensure high reproducibility, and it is referred to as a pipeline.There is a wealth of details, which I presume will help replication of the pipeline and I expect that this is the main interest of the paper. Overall the paper describes a near industrial approach to obtaining high fidelity mapping in a highly systematic fashion. The validity of the paper is not addressed.

The paper exposes a methodology and approach to mapping whole-brain connectivity in a primate species, and presents a data-resource generated by the methodology. As such this is not a hypothesis-driven study. Nevertheless, we do address questions regarding the validity of the approach. The reviewer is right to highlight issues of individual variability, which we have addressed more extensively in added appendix sections as well as text added to the body of the manuscript.

Contrary to what is suggested in the Abstract where it states that 'we are able to overcome individual variation exhibited by marmosets to obtain routine and high quality maps to a common atlas framework'. This misleading sentence could suggest that brain variability in terms of areal layouts or connectivity have been addressed in the study. This is not the case.

We apologize for the unclear phrasing in the Abstract, which we have now rectified. What we had meant to say (and which is now more explicitly stated), that we are able to ameliorate the issue of individual variations in brain anatomy while placing tracer injections by using in-vivo MRI and mapping to a reference atlas to target specific brain compartments in a given marmoset. This is an important methodological difference from the same approach previously applied to the mouse, since in the latter case it is not standard procedure to use in-vivo MRI to guide tracer injection placement. Further, diffeomorphic mapping applied to the combined in-vivo, ex-vivo MRI and 3D multiple histological series, allows us to determine the injection and projection locations a-posteriori using computational means. These issues are further discussed in the added appendix sections.

The paper is not a study in the usual sense, nevertheless it is subdivided into Introduction, Materials and methods, Results and Discussion. None of these elements have a common bearing. For instance the Introduction has a long section on the phylogenetic tree, which might lead one to expect that the paper deals with marmosets. It doesn't.

We respectfully disagree with the reviewer that the manuscript does not deal with marmosets. Indeed, it is focused on mapping brain-wide connectivity in the marmoset and presents the largest data set available to date of 3D brain-wide histological series in the marmoset to address questions of brain connectivity. While we do not have complete coverage, and it will take some time to fully analyze the implications of the data gathered, we feel it valuable at this time to share the experience we have gathered with the marmoset community, as well as the data resource. Parenthetically we should note that the resource has already been well received and is already being used by researchers wishing to know about the existence of projections between different brain regions in the marmoset.

Marmoset brains are processed in the pipeline but we learn nothing about them that would warrant understanding their position in evolution.

We agree that we do not yet have new knowledge to add to the evolution of the marmoset brain (we hope to extract such knowledge in future analysis of the data gathered). However, we do feel it appropriate to insert a phylogenetic discussion to contextualize brain-wide connectivity mapping, as this activity has so far been focused on the mouse. We also feel the need to re-invigorate a comparative discussion in the context of modern approaches to neuroanatomy and feel that comparative/evolutionary discussions might become a more standard component of the background portions of papers that might otherwise simply be technique focused. This would give a better understanding of the utility of these techniques beyond say transgenic mice. Even the simple size difference in the brains across taxa may make some of the techniques less compelling than others, and this is indeed a comparative question.

Likewise, the Introduction touches on curated databases and develop at length the Felleman and Van Essen, 1991 study leading to the work that established the CoCoMac data base. This might lead one to expect that the paper would lead to data and that the merits and otherwise of curated data would be relevant. That neither is the case.

Our goal in reviewing the literature was not to present a comparative study of the findings of Van Essen et al. with a connectivity map in the marmoset. Such a comparison will be appropriate to make in the future once our data is more fully analyzed. Rather, the comparison we wish to draw at present is methodological, between connectivity matrices generated largely by visual inspection of raw materials that are not available to the reader in any image format, and presented as tabular summaries – and our present approach of gathering whole-brain histological image data which is open to the reader for examination. Secondly, we wished to examine the number of tracer injections placed in the past in the literature studies, as a comparison point for the current study. This is presented in tabular form.

My point is that the Introduction could be scrapped with no loss to the paper; Idem for the Discussion. If this paper were to be a study, various parameters of the pipeline would be modulated so as to see how this impacts the results.

It is impractical to repeat the entire study with different parameters. However, we did consider different pipeline parameters when designing the study, and some of these considerations are now detailed in the appendix sections (9 and 10). Rather than remove the introductory and Discussion sections, we have expanded them to include discussions on the grid-injection approach, individual variations, and planning for completion of whole-brain coverage.

For example, if one is to embark on a grid approach to determining the connectome (as is the case here) what is the optimal frequency to be used? Clearly this is related to brain size.

A discussion of the spatial grid was previously included, but we have expanded on it.

If one is to use a template approach, as in the present study, what impact does that have on the design of the pipeline and what are the controls that would allow one to isolate the template effect? What are the short-comings of a grid approach? What are the short-comings of a template approach? These are burning issues facing connectomics today that are not touched upon by this methodological paper.

We have included a more comprehensive discussion of the grid approach, discussing several of the issues raised by the reviewers in this context. By a “template approach” we presume the reviewer refers here to placement of injections based on a reference atlas. The approach utilized in the study is a hybrid, where the grid is adapted so as to avoid major anatomical boundaries. We discuss the context of such a hybrid approach.

Reviewer #2:Achieving a comprehensive connectivity map of the primate brain at the 'mesoscale' is a challenge and the authors are to be commended for taking it on. The manuscript is essentially the description of a method that the authors have designed and tested for its feasibility. It requires high throughput, both experimentally and computationally. The manuscript has indeed described a feasible way of achieving the aim of a whole-brain connectivity map of the marmoset. This can potentially pave the way for applying the same techniques to larger brains such as the macaque's.I see three major challenges in the ultimate objective for producing a useful connectivity map for the marmoset:1) How well can the size of injections be controlled? Spread to neighbouring areas will be a problem. However, with greater number of animals and appropriate computational tools, I agree that one can improve the results.

We have included an analysis of the injection sizes in the data set acquired so far. We place pressure injections, so that the injection size is nominally controlled by the volume injected, but examination of the data shows that there is additional variability, some of which may originate from differing fluid transport properties in different brain regions. We also analyze the data set gathered so far to estimate the number of injections that spread across anatomical boundaries (we find that most of the injections are localized within major compartments).

2) Smaller the site of injection, better will be the final resolution of the connectivity map. However, that will increase the number of injections that need to be made.

The reviewer is quite correct in noting the relation between the size of the injection and the resolution of the connectivity map. This is ultimately a practical issue as the number of injections is upper bounded by the availability of experimental animals. Some guidance as to size of injections may be obtained from the distribution of sizes of the neuroanatomical compartments. While we capture the major compartments, it is not practical to inject the smallest compartments delineated on the reference atlas in question. We therefore chose a pragmatic cutoff.

3) There will be animal-to-animal variability and with the marmoset, I believe one does not have the same degree of luxury of having precisely the same genetic strain as with mice. This might require the experiments to be repeated with a large number of animals. I am not sure that "access to high-quality in-vivo and ex-vivo MRI" as stated in the manuscript will be adequate.I would have liked to have seen some discussion about realistic numbers of animals and injections and the time needed for the project to be completed. I guess this would require a large consortium of laboratories. Some plausible projections would add to the feasibility of this ambitious programme.

Indeed, we cannot constrain to an inbred strain in the present study as such a strain is not available. No previous anatomical study has been done with inbred marmosets, so this might be too high a bar for the current study. As for the utility of in-vivo and ex-vivo MRI: as discussed elsewhere, this is for individually-targeted injection placement, and a-posteriori analysis of individual variation.

We have now provided a more detailed projection of the number of animals and injections required to obtain the comprehensive brain-wide connectivity map in the marmoset.

Reviewer #3:This is an important, relevant, and well carried out methodological contribution. While systematic projectional analyses of mouse brains have become available since 2014, the application and extension of such approaches to larger brains, and most notably a non-human primate brain, is of substantial relevance for obtaining comparative data on large-scale brain connectivity. I particularly commend the authors on Figures 1 (evolutionary context) and Figure 7 (detailed explanation and illustration of the connectivity matrix generation).Importantly, the data viewer should be operational as advertised. Currently, http://riken.marmoset.brainarchitecture.org/ doesn't load, and shows black data only. Please assure that the data browser is operational at review stage, since this is one important aspect of the presented methods.

We do apologize that the reviewer found the data portal to be offline, possibly during an update while the manuscript was in review. The portal is now online and we do not anticipate significant downtime, except possibly for brief outages during maintenance. We have simplified the web portal address to http://marmoset.brainarchitecture.org/